# Transcription-independent regulation of STING activation and innate immune responses by IRF8 in monocytes

Wei-Wei Luo[1], Zhen Tong[1], Pan Cao[2], Fu-Bing Wang[3], Ying Liu[2], Zhou-Qin Zheng[1], Su-Yun Wang[1], Shu Li ®[2] & Yan-Yi Wang ®[1] ✉

Sensing of cytosolic DNA of microbial or cellular/mitochondrial origin by cGAS initiates innate immune responses via the adaptor protein STING. It remains unresolved how the activity of STING is balanced between a productive innate immune response and induction of autoimmunity. Here we show that interferon regulatory factor 8 (IRF8) is essential for efficient activation of STING-mediated innate immune responses in monocytes. This function of IRF8 is independent of its transcriptional role in monocyte differentiation. In uninfected cells, IRF8 remains inactive via sequestration of its IRF-associated domain by its N- and C-terminal tails, which reduces its association with STING. Upon triggering the DNA sensing pathway, IRF8 is phosphorylated at Serine 151 to allow its association with STING via the IRF-associated domain. This is essential for STING polymerization and TBK1-mediated STING and IRF3 phosphorylation. Consistently, IRF8-deficiency impairs host defense against the DNA virus HSV-1, and blocks DNA damage-induced cellular senescence. Bone marrow-derived mononuclear cells which have an autoimmune phenotype due to deficiency of Trex1, respond to IRF-8 deletion with reduced pro-inflammatory cytokine production. Peripheral blood mononuclear cells from systemic lupus erythematosus patients are characterized by elevated phosphorylation of IRF8 at the same Serine residue we find to be important in STING activation, and in these cells STING is hyper-active. Taken together, the transcription-independent function of IRF8 we describe here appears to mediate STING activation and represents an important regulatory step in the cGAS/STING innate immune pathway in monocytes.

Sensing of cytosolic DNA derived from microbial infection or dislocated nuclear and mitochondrial DNA is important for host defense against DNA pathogens and certain cellular abnormalities[1–3]. The Cyclic GMP-AMP synthase (cGAS) is a widely used sensor of cytosolic double-strand (ds) DNA. Upon binding to dsDNA, cGAS utilizes GTP and ATP as substrates to synthesize the second messenger cyclic GMP-AMP (cGAMP)[4–6]. cGAMP binds to the ER-associated adaptor protein STING (also known as MITA, ERIS and MPYS)[7–10], which induces its polymerization and cellular trafficking. During these processes, the serine/threonine kinase TBK1 is recruited to STING and activated by

[1]Key Laboratory of Special Pathogens and Biosafety, Wuhan Institute of Virology, Center for Biosafety Mega-Science, Chinese Academy of Sciences, Wuhan 430071, China. [2]Department of Infectious Diseases, Zhongnan Hospital of Wuhan University, Frontier Science Center for Immunology and Metabolism, Medical Research Institute, Wuhan University, Wuhan 430071, China. [3]Department of Laboratory Medicine, Zhongnan Hospital of Wuhan University, Wuhan University, Wuhan 430071, China. ✉e-mail: wangyy@wh.iov.cn

induced proximity. Phosphorylation of STING at its pLxIS motif by TBK1 increases their binding affinity and facilitates further recruitment and activation of TBK1[11]. Phosphorylation of STING also causes recruitment of the transcription factor Interferon (IFN) Response Factor 3 (IRF3), which is then phosphorylated at its pLxIS motif by TBK1. Phosphorylated IRF3 dissociates from the STING complex and translocates to the nucleus to induce transcription of innate immune effector genes, such as type I IFNs and proinflammatory cytokines[3,11–15]. In addition to critical roles in innate immune responses to microbial pathogens, the cGAS-STING pathway is also involved in cellular senescence[16,17] and anti-tumor immunity[18,19].

Since STING plays critical roles in innate immune responses to DNA pathogens and aberrant self DNA, dys-regulation of STING has been implicated in various autoimmune syndromes such as systemic lupus erythematosus (SLE), multiple sclerosis, Aicardi–Goutières syndrome and STING-associated vasculopathy with onset in infancy (SAVI)[20–24]. Hyperactivity of STING-mediated signaling also contributes to acute inflammation in myocardial infarction and chronic inflammation in liver diseases and pancreatitis[25,26]. How STING activity is properly regulated for efficient host defense while inert for autoimmunity remains enigmatic.

The IRF family members are transcription factors that are critically involved in innate immune responses and the development of immune cells[27]. Among IRFs, IRF3 and IRF7 play crucial roles in induction of type I IFNs and other antiviral effectors after viral infection. IRF3 is mainly involved in initiation of IFN-β expression at the early phase, whereas IRF7 facilitates rapid and large-scale IFN-α/β production at the late phase[28,29]. IRF8, also known as IFN consensus sequence binding protein (ICSBP), contains a DNA binding domain (DBD) at its N-terminus and a middle IRF association domain (IAD). In general, IRF8 binds to specific DNA sequences when it forms heterodimers with partner transcription factors via its IAD and thereby acts either as an activator or a repressor to facilitate the transcription of downstream genes[30–34]. In other cases, IRF8 has been reported to directly regulate the transcription of several genes[35–37]. IRF8 is an important regulator for macrophage, dendritic cell (DC) and B-cell development and implicated in Th17, Th9, and Treg cell differentiation[38–42]. Moreover, IRF8 has been reported to regulate the adaptive natural killer cell response and inflammasome activation[35,36]. Previous studies have also demonstrated that IRF8 amplifies the second phase of type I IFN induction in DCs by sustaining RNA polymerase II recruitment to the IFN promoters in responding to infections of murine cytomegalovirus (MCMV) and Newcastle disease viruses (NDV)[43]. *Irf8*[−/−] mice are more susceptible to lethal vaccinia virus (VACV) and lymphocytic choriomeningitis virus (LCMV) infection, but have an intact antiviral response to vesicular stomatitis virus (VSV)[44]. These studies suggest that IRF8 is differentially involved in host defense against different types of viruses. Genome-wide association studies (GWAS) demonstrate that sequence variants in *IRF8* gene are significant risk factors for multiple chronic inflammatory diseases in humans[45]. How IRF8 is differentially involved in innate immune and inflammatory responses is unknown.

In this study, we show that IRF8 plays a transcription-independent and phosphorylation-dependent role in STING-mediated innate immune response, which provides added to our understanding of how innate immune responses to cytosolic DNA in monocytic cells are properly regulated for efficient and balanced host defense.

## Results

### IRF8 is essential for DNA but not RNA virus-triggered transcription of antiviral genes in monocytes

To investigate potential roles of IRF8 in innate antiviral responses, we examined the effects of IRF8-deficiency on induction of downstream antiviral genes triggered by different types of viruses. We generated IRF8-deficient human monocytic THP1 cells by the CRISPR-Cas9 method. qPCR experiments indicated that transcription of

downstream antiviral effector genes (*IFNB1* and *IFIT1*) induced by the DNA virus herpes simplex virus 1 (HSV-1) was markedly decreased in IRF8-deficient THP1 cells in comparison to control cells. In contrast, IRF8-deficiency had no marked effects on transcription of downstream antiviral genes induced by the RNA virus Sendai virus (SeV) (Fig. 1a). Viral nucleic acids are major pathogen-associated molecular pattern (PAMP) recognized by cytoplasmic innate immune sensors. IRF8-deficiency inhibited transcription of *IFNB1*, *IFIT1,* and *CXCL10* genes triggered by transfected synthetic DNA, including 120-mers dsDNA representing the genome of HSV-1 (HSV120), 70-mers dsDNA representing the genome of VACV (VACA70), dsDNA of approximately 90 bp (dsDNA90), and interferon stimulatory DNA(ISD)[46] (Fig. 1b). In these experiments, IRF8-deficiency had no marked effects on transcription of downstream antiviral genes induced by transfected low-molecular weight (LMW) poly(I:C) (which is an RNA analog sensed by RIG-I) (Fig. 1b). These results suggest that IRF8 plays an important role in viral DNA- but not RNA-triggered induction of downstream antiviral genes.

We next investigated the roles of murine *Irf8* in innate antiviral responses (in this manuscript, to avoid confusion of murine and human proteins, only the first letter in the name of a murine protein is capitalized). We compared expression of downstream antiviral genes induced by different types of viruses in wild-type (WT), *Irf3*[−/−] and *Irf8*[−/−] murine bone marrow-derived macrophage (BMDM). The results indicated that Irf3-deficiency inhibited transcription of *Ifnb1* and *Cxcl10* genes induced by all examined viruses, including the RNA viruses SeV, VSV and NDV, and the DNA viruses HSV-1 and VACV, whereas Irf8-deficiency specifically inhibited transcription of downstream genes induced by the DNA viruses HSV-1 and VACV but not the RNA viruses SeV, VSV and NDV (Fig. 1c). Consistently, Irf3-deficiency inhibited secretion of Ifn-β induced by all the examined viruses, whereas Irf8-deficiency inhibited Ifn-β production induced by the examined DNA but not RNA viruses (Fig. 1d). RNA-seq analysis indicated that global transcription of type I IFN genes and interferon-stimulated gene (ISG) induced by HSV-1 was impaired in Irf8-deficient compared with WT BMDMs (Fig. 1e). The Gene Set Enrichment Analysis (GSEA) showed that Irf8-deficiency had no significant effects on global alteration of genes related to cytosolic DNA-sensing pathway in unstimulated cells (Fig. S1a), which suggests that role of Irf8 in HSV-1-induced transcription of downstream genes is not through its effects on basal expression of cytosolic DNA-sensing pathway genes. Irf8-deficiency also inhibited transcriptions of *Ifnb1* and *Cxcl10* genes (Fig. 1f) and secretion of Ifn-β (Fig. 1g) induced by transfected herring testis (HT)-DNA but not LMW-poly (I:C). These results suggest that Irf8 is required for innate immune responses to DNA but not RNA viruses, while Irf3 is important for innate immune responses to both types of viruses in BMDMs.

### Irf8 is essential for DNA-triggered activation of the Sting-Irf3 axis

Previously, it has been shown that Irf3 acts as a critical transcription factor in innate antiviral immune responses[29]. To determine whether Irf8 also acts as a transcription factor and its relationship with Irf3 in innate antiviral signaling cascades, we dissected virus-triggered signaling cascades in *Irf8*[−/−] and *Irf3*[−/−] BMDMs. As a downstream transcription factor, Irf3-deficiency had no marked effects on HSV-1 or VACV-induced phosphorylation of Sting[S365] and Tbk1[S172] in BMDMs, which are hallmarks of activation of these upstream components[14] (Fig. 2a). In these experiments, Irf3-deficiency also had no marked effects on SeV-induced Tbk1[S172] phosphorylation (Fig. S1a). In contrast, Irf8-deficiency impaired HSV-1 or VACV-induced phosphorylation of Sting[S365] and Irf3[S388] but not Tbk1[S172] in BMDMs (Fig. 2a). Irf8-deficiency also impaired HSV-1-induced translocation of Irf3 to the nucleus in BMDMs (Fig. 2b). However, Irf8-deficiency had no marked effects on SeV-induced phosphorylation of Tbk1[S172] and Irf3[S388] in BMDMs (Fig. S1b). These results suggest that Irf8 is required for Tbk1-mediated

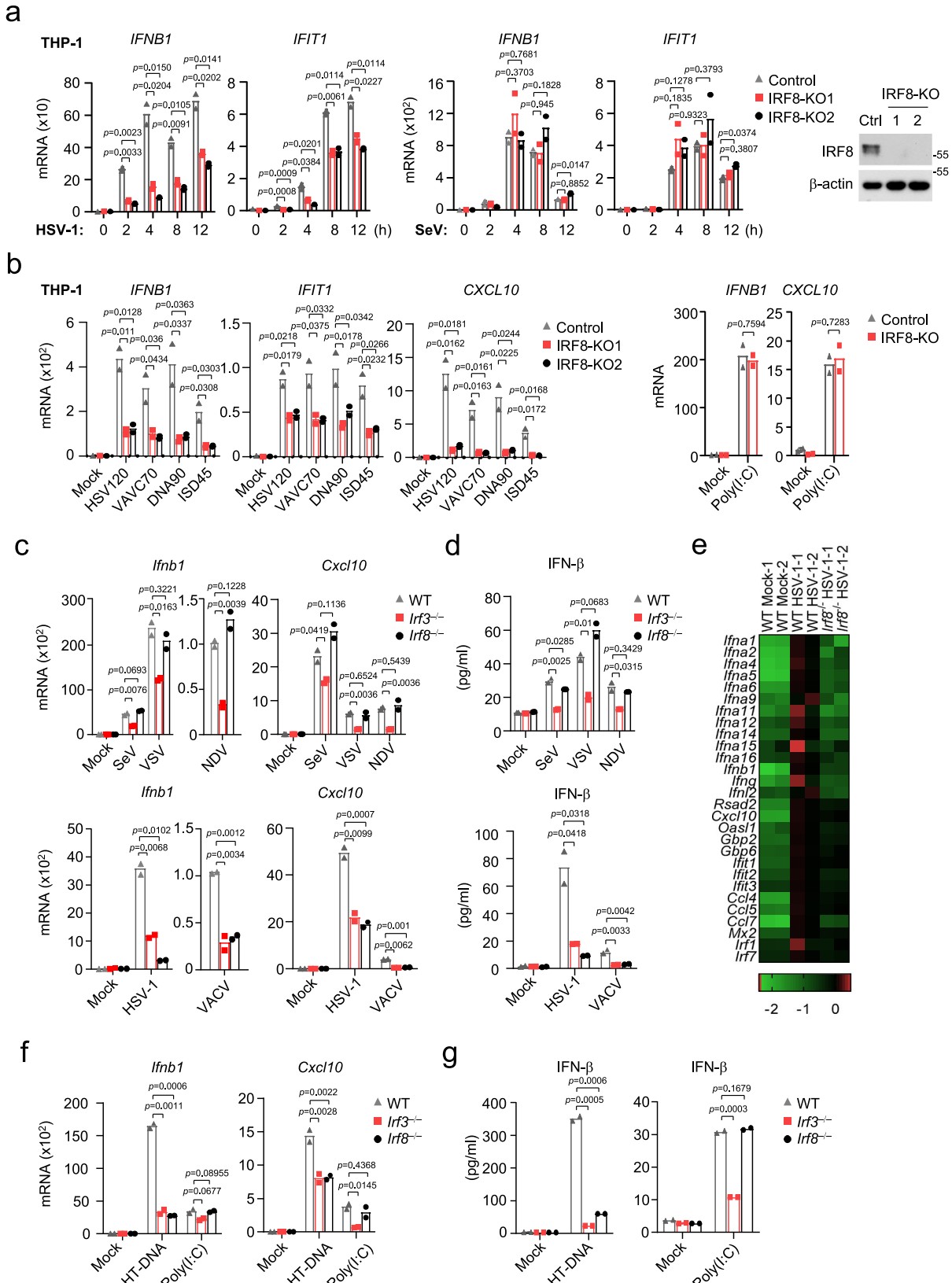

Sting phosphorylation and its activation of Irf3 following DNA virus infection in BMDMs. Notably, Irf8-deficiency also obviously enhanced virus-triggered phosphorylation of Ikkβ, p65 and Ikbα in BMDMs (Fig. 2a and Fig. S1b). Similarly, Irf8-deficiency impaired HSV-1-induced phosphorylation of Sting$^{S365}$ and Irf3$^{S388}$ but not SeV-induced phosphorylation of Tbk1$^{S172}$ and Irf3$^{S388}$ in bone marrow-derived dendritic cell (BMDC) (Fig. S1c), suggesting that Irf8 plays a conserved function in DNA virus-induced activation of Sting-Irf3 axis in different immune cells. Recent studies have reported that Sting-mediated autophagy does not depend on phosphorylation of its S365[47]. We found that Irf8-deficiency impaired cGAMP-induced phosphorylation of Sting$^{S365}$ and Irf3$^{S388}$ in BMDMs but had no effects on cGAMP-induced LC3

**Fig. 1 | IRF8-deficiency inhibits DNA-triggered signaling in monocytes. a** qPCR analysis of *IFNB1* and *IFIT1* mRNAs in IRF8 knockout (IRF8-KO-1 and IRF8-KO-2) or control THP-1 cells (transduced with the lenti-CRISPR-V2 containing a control gRNA) infected with SeV or HSV-1 for the indicated times. The knockout efficiencies of IRF8 were shown by immunoblots. **b** qPCR analysis of mRNA levels of the indicated genes in IRF8-KO or control THP-1 cells transfected with the indicated nucleic acids for 4 h. qPCR analysis of *Ifnb1* and *Cxcl10* mRNAs (**c**) and ELISA analysis of Ifn-β secretion (**d**) in WT, *Irf8*⁻/⁻ and *Irf3*⁻/⁻ BMDMs left uninfected or infected with the indicated viruses for 4 h. **e** Heat map of hierarchical clustering analysis of RNA-sequencing data from WT and *Irf8*⁻/⁻ BMDMs left uninfected or infected with HSV-1 for 4 h. The scale (log10) represents the changes in gene expression among different samples. qPCR analysis of *Ifnb1* and *Cxcl10* mRNAs (**f**) and ELISA analysis of Ifn-β secretion (**g**) in WT, *Irf8*⁻/⁻ and *Irf3*⁻/⁻ BMDMs transfected with the indicated nucleic acids for 4 h. Data in **a**–**d**, **f** and **g** are presented as mean. *n* = 2 technical replicates. Data were analyzed by unpaired two-tailed Student's *t*-test. Source data are provided as a Source data file.

conversion, which is a marker for autophagy (Fig. S1d). These results suggest that Irf8 is important for Sting-mediated Irf3 activation but dispensable for Sting-mediated NF-κB activation and autophagy.

It has been shown previously that Irf8 is essential for the development of plasmacytoid dendritic cell (pDC) and type 1 conventional dendritic cell (cDC1) but not type 2 conventional dendritic cell (cDC2)[48]. We investigated whether the effects of Irf8 on innate antiviral responses are secondary to its involvement in differentiation of DCs. We isolated bone marrow cells from WT mice and cultured these cells with murine Flt3L for 8 days for differentiation of DCs. Knockout of Irf8 was then performed by the CRISPR-Cas9 method in these differentiated BMDCs. qPCR experiments showed that knockout of Irf8 impaired HSV-1-induced transcription of *Ifnb*, *Il6*, *Ifit1*, and *Cxcl10* genes in Flt3L-BMDCs (Fig. S2a). We further sorted the pDC (B220⁺), cDC1 (Bst2⁻ B220⁻ CD11c⁺ MHCII⁺ CD24⁺ CD172a⁻) and cDC2 (Bst2⁻ B220⁻ CD11c⁺ MHCII⁺ CD172a⁺) cells from Flt3L -BMDCs (Fig. S2b). We found that Irf8-deficiency markedly inhibited HSV-1-induced transcription of downstream antiviral genes (Fig. S2c) and production of Ifn-β and IP-10 (Fig. S2d) in pDCs and cDC1s but not cDC2s. qPCR experiments showed that *Irf8* mRNA was detected in pDCs and cDC1s but not cDC2s (Fig. S2e). It has been shown that TLR9, which is expressed in DCs, also senses viral DNA in response to HSV-1 infection[49]. We therefore also stimulated the cells with the murine Sting-specific agonist DMXAA, which is independent of the TLR9 signaling. The results indicated that Irf8-deficiency also markedly inhibited DMXAA-induced transcription of downstream antiviral genes (Fig. S2c) and production of Ifn-β and IP-10 (Fig. S2d) in pDCs and cDC1s. Collectively, these results suggest that Irf8 plays an essential role in Sting-mediated innate immune responses in Irf8-expressing macrophages and DCs, which is independent of its roles in differentiation of these cells.

Previously, it has been shown that Irf8 is a transcription factor which regulates monocyte and DC development[45]. We next investigated whether the transcriptional activity of Irf8 contributes to its regulation of innate immune responses to DNA. *Irf8*⁻/⁻ BMDMs were reconstituted with WT Irf8, Irf8(T80A) and Irf8(K79E), two mutants that lack transcriptional activities[50,51]. We found that HSV-1-induced transcription of *Ifnb1* and *Cxcl10* genes (Fig. 2c) and secretion of Ifn-β and IP-10 (Fig. 2d) in *Irf8*⁻/⁻ BMDMs were fully rescued by reconstitution with WT as well as the two Irf8 mutants. In these experiments, reconstitution with WT but not two Irf8 mutants in *Irf8*⁻/⁻ BMDMs rescued LPS/IFN-γ-induced transcription of *Il12b* gene, suggesting that these Irf8 mutants lost their transcriptional activity (Fig. 2c). These results suggest that the roles of Irf8 in innate immune responses to DNA viruses are independent of its transcriptional activity.

## Irf8 mediates polymerization of Sting and its recruitment of Irf3

We next investigated the mechanisms on how Irf8 regulates Sting-mediated innate antiviral responses. It has been reported that Irf8 is located both in the cytoplasm and nucleus, and it is induced and translocated to the nucleus in response to stimulation by LPS or IFN-γ[52]. Immunofluorescent staining indicated that Irf8 was localized both in the nucleus and cytoplasm in rest BMDMs, and HSV-1 infection for 6 h increased the amounts of Irf8 in the cytoplasm (Fig. 3a). Subcellular fractionation analysis confirmed that the level of Irf8 was increased in the cytosolic fraction after HSV-1 infection (Fig. 3b). These results

suggest that viral infection induces increase of Irf8 in the cytoplasm, which may contribute to its cytoplasmic regulation of innate antiviral responses. Consistent with a transcription-independent function of Irf8 in the cytoplasm, we found that cGAMP-induced phosphorylation of human STING^S366 (corresponding to mouse Sting^S365) and IRF3^S396 (corresponding to mouse Irf3^S388) (Fig. 3c), as well as transcription of downstream antiviral genes (Fig. 3d), was impaired in IRF8-deficient human monocytic THP1 cells. In these cells, IRF8-deficiency had no effects on the expression of cGAS, suggesting that the effects of IRF8-deficiency is not dependent on the upstream cGAS (Fig. S3a). In light of our above data, which suggests that Irf8 is essential for DNA virus-triggered phosphorylation of Sting^S365 and Irf3^S388 but not Tbk1^S172 in BMDMs (Fig. 2a and Fig. S1), these data together suggest that IRF8 acts at the STING-TBK1 level downstream of cGAMP. Transient transfection and co-immunoprecipitation experiments indicated that IRF8 was associated with STING but not cGAS, TBK1 or IRF3 (Fig. 3e). Endogenous co-immunoprecipitation experiments indicated that Irf8 barely associated with Sting in un-stimulated cells, but their association was markedly increased following HSV-1 infection for 2 h (Fig. 3f). Confocal microscopy indicated that in rest cells, IRF8 was mostly co-localized with STING in the endoplasmic reticulum (ER) and partially in the nucleus. Following cGAMP stimulation, IRF8 was decreased in the nucleus and formed cytoplasmic punctate structures co-localized with the ER-Golgi intermediate compartment (ERGIC) and Golgi (Fig. 3g). In addition, confocal microscopy also indicated that Irf8 was co-localized with S365-phosphorylated Sting at the punctate structures following HSV-1 infection or DNA transfection or cGAMP stimulation (Fig. 3h). Gel filtration experiments indicated that HSV-1 infection of BMDMs induced shift of Irf8, Sting and Tbk1 from lower to higher molecular weight fractions, in which these proteins overlapped (Fig. S3b). It is noted that phosphorylation of Sting^S365 was mostly detected in the Irf8-overlapping high molecular weight fractions following HSV-1 infection (Fig. S3b). Taken together, these results suggest that Irf8 is associated with Sting, and this association is important for Sting-mediated Irf3 activation after DNA virus infection.

IRF8 contains a DNA binding domain (DBD) at its N-terminus and a middle IAD. Domain mapping experiments indicated that the IAD but not DBD of IRF8 is required and sufficient for its interaction with STING, whereas a fragment (aa190-221) of STING is required for its interaction with IRF8 (Fig. S4a). It has been shown that aa191-221 of STING is located in its ligand (cGAMP)-binding domain (LBD), which is important for cGAMP-induced conformational changes and formation of polymer interface of STING and its subsequent activation[11,12,53]. Therefore, we investigated whether IRF8 regulates STING polymerization. Native-PAGE indicated that basal dimerization as well as HSV-1- or cGAMP-induced dimerization of Sting was comparable between *Irf8*⁻/⁻ and wild-type BMDMs (Fig. 4a). However, HSV-1-induced Sting polymerization as well as Sting^S365 phosphorylation were abrogated in *Irf8*⁻/⁻ comparing to wild-type BMDMs (Fig. 4b). In these experiments, HSV-1-induced Tbk1 polymerization was also moderately decreased in *Irf8*⁻/⁻ BMDMs (Fig. 4b). In addition, HSV-1-induced shift of Sting and Tbk1 to higher molecular weight complexes as well as Sting^S365 phosphorylation was impaired in Irf8-deficient BMDMs (Fig. 4c). Interestingly, HSV-1-induced recruitment of Irf3 to Sting was impaired in Irf8-deficient BMDMs, but the recruitment of Tbk1 to Sting was barely affected (Fig. 4d). These results suggest that Irf8 is not

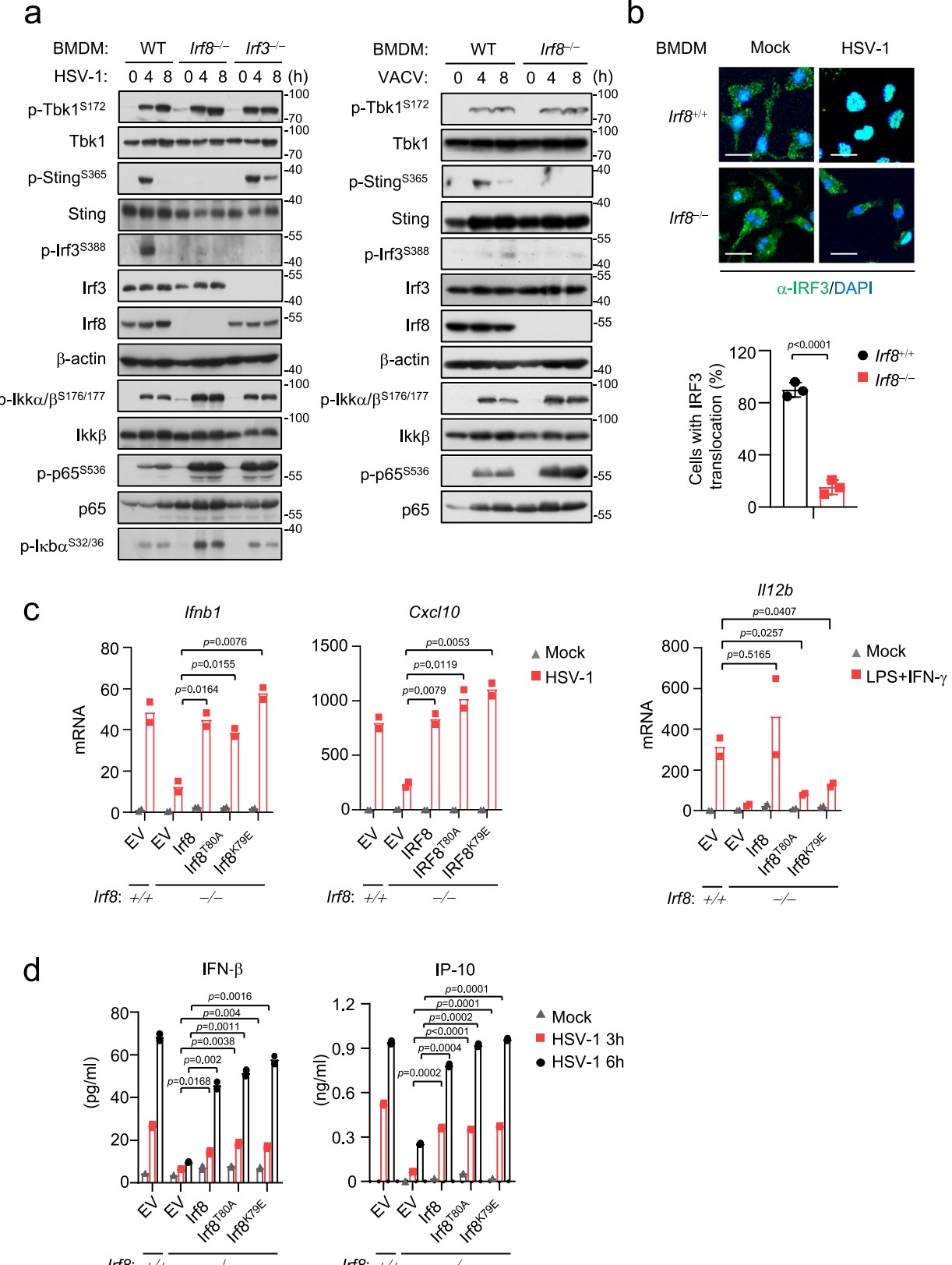

required for Sting dimerization and its recruitment of Tbk1, but essential for its polymerization and recruitment of Irf3 after DNA virus infection. Previously, it has been demonstrated that STING exists in an inactive form because of sequestration of the LBD by its C-terminal tail (CTT), which blocks its polymerization interface[12,54]. Consistently, overexpression of IRF8 impaired the interaction of STING with its CTT

(Fig. S4a), suggesting that interaction of IRF8 with STING promotes release of its inhibitory CTT and subsequent polymerization. Interestingly, we noted in the domain-mapping experiments that the association of IRF8 with STING was enhanced when the N- or C-terminus of IRF8 was deleted (Fig. S4b). Coimmunoprecipitation experiments indicated that the N-terminal DBD (aa1-200) and CTT

**Fig. 2 | Irf8 mediates activation of Sting-Irf3 axis independent of its transcriptional activity. a** Immunoblot analysis of the indicated proteins in WT, *Irf8⁻/⁻* or *Irf3⁻/⁻* BMDMs un-infected or infected with HSV-1 (left) or VACV (right) for the indicated times. **b** Immunostaining showing Irf3 (green) in wild-type or *Irf8⁻/⁻* BMDMs infected with HSV-1 for 6 h. Scale bars, 100 μm. The number of cells with Irf3 translocation was counted from three random fields (lower histogram). Graph shows mean ± SEM, *n* = 3 independent samples. Data were analyzed by unpaired two-tailed Student's t-test. **c** *Irf8⁻/⁻* BMDMs were reconstituted with wild-type Irf8 or

the indicated mutants. The cells were infected with HSV-1, or treated with LPS (10 ng/mL) plus IFN-γ (50 ng/mL) for 6 h before qPCR analysis of the mRNA levels of the indicated genes. **d** *Irf8⁻/⁻* BMDMs were reconstituted with wild-type Irf8 or the indicated mutants. The cells were infected with HSV-1 for the indicated times, and then ELISA analysis of Ifn-β and IP-10 secretion was performed. Data in **c**, **d** are presented as mean, *n* = 2 biological replicates. Data were analyzed by unpaired two-tailed Student's t-test. Source data are provided as a Source data file.

(aa375-426) of IRF8 interacted with its middle IAD (Fig. S4c). Overexpression of the DBD or CTT of IRF8 disrupted the interaction of full-length IRF8 with STING (Fig. S4d). These results suggest that IRF8 may also exist in an auto-inhibitory status in rest cells. Consistently, HSV-1-induced transcription of *Ifnb1* and *Cxcl10* genes (Fig. S5a) and production of Ifn-β and Cxcl10 (Fig. S5b) in *Irf8⁻/⁻* BMDMs were rescued by reconstitution with WT Irf8 and its IAD-containing (Irf8ΔDBD) but not IAD deletion (Irf8ΔIAD) mutant. Whereas overexpression of Irf8 DBD or CTT inhibited activation of the *IFNB1* promoter induced by cGAMP stimulation (Fig. S5c).

### Phosphorylation of Irf8^S151 is essential for Sting-mediated signaling

Since Irf8 is important for Tbk1-mediated phosphorylation of Sting and Irf3 in monocytic cells, we next investigated whether the functions of Irf8 in innate antiviral responses are regulated by phosphorylation. Sequence analysis predicted 7 serine residues in Irf8 which are conserved in mammals. We individually mutated these serine residues to alanine or the phosphorylation mimic aspartic acid. Reporter assays indicated that the S151A but not the other mutants lost the ability to synergize with Sting to activate the *IFNB1* promoter (Fig. 5a). IRF^S151A also had a dramatically decreased ability to interact with Sting (Fig. 5b). cGAMP stimulation enhanced the association of STING with wild-type IRF8 but not IRF8^S151A (Fig. 5c). Reconstitution of *Irf8⁻/⁻* BMDMs with wild-type Irf8, Irf8^S151D but not Irf8^S151A rescued HSV-1-induced production of Ifn-β and Cxcl10 (Fig. 5d). Consistently, wild-type Irf8 but not Irf8^S151A rescued HSV-1-induced phosphorylation of Sting^S365 in *Irf8⁻/⁻* BMDMs (Fig. 5e). In these experiments, HSV-1 also failed to induce phosphorylation of the Irf8^S151A (Fig. 5e). Since S151 of IRF88 is located in its DBD which sequesters its IAD domain and impairs its association with Sting, we next determined whether phosphorylation of IRF8^S151 triggers the release of its IAD. We found that the IRF8 S151D mutation, which mimics its constitutive phosphorylation, markedly inhibited the interaction of DBD with IAD of IRF8 (Fig. 5f). Collectively, these results suggest that phosphorylation of IRF8^S151 releases its auto-inhibition and promotes its interaction with and activation of STING. We next investigated whether IRF8 is phosphorylated by TBK1. We found that Tbk1-deficiency or its inhibitor BX795 had no marked effects on HSV-1-induced Irf8 phosphorylation (Fig. 5g, h), suggesting that Tbk1 is not a kinase for Irf8.

### Irf8 is essential for innate immunity to DNA virus in vivo

We next investigated whether Irf8 is required for host defense in vivo. We infected 8-week old wild-type and Irf8-deficient mice with HSV-1 intravenously (i.v.). All infected *Irf8⁻/⁻* mice developed discrepant lethargy and ataxia within 5 days of HSV-1 infection and died within 2 days after the appearance of symptoms (Fig. 6a). In these experiments, 70% of the infected wild-type mice exhibited the symptoms, which died over 7–10 days after the appearance of symptoms (Fig. 6a). The sera from *Irf8⁻/⁻* mice infected with HSV-1 had significantly lower levels of Ifn-β and Cxcl10 compared with those from wild-type mice (Fig. 6b). The mRNA levels of *Ifnb1* and *Cxcl10* genes were severely impaired in lungs and livers from *Irf8⁻/⁻* compared with wild-type mice at 24 h after HSV-1 infection (Fig. 6c). Plaque assays confirmed that Irf8-deficiency resulted in increased HSV-1 titers in the lungs and livers from *Irf8⁻/⁻* mice compared with wild-type controls 72 h after infection

(Fig. 6d). HSV-1 is a neurotropic virus and the leading cause of sporadic viral encephalitis[55]. We found that the mRNA levels of *Ifnb1* and *Cxcl10* genes were significantly lower whereas the viral loads were significantly higher in the brains of *Irf8⁻/⁻* than wild-type mice 5 days post HSV-1 infection (Fig. 6e, f). Collectively, these data suggest that Irf8 plays an important role in host defense against HSV-1 infection in vivo.

### Irf8 is important for damage-induced senescence

Given an emerging role of cGAS-STING axis in cellular senescence[16,17], we also examined the potential roles of IRF8 in senescence. We found that Irf8-deficiency prevented the senescence phenotypes induced by DNA-damage inducers such as hydroxyurea (HU) and mitomycin C (MMC) in BMDMs, as revealed by senescence-associated β-galactosidase (SA-β-Gal) staining (Fig. S6a). Another hallmark of senescent cells is senescence-associated secretory phenotype (SASP), which can be assessed by measuring the expression levels of SASP genes[56]. qPCR analysis indicated that mRNA levels of the SASP genes, including *Il1b*, *Il8*, *Il6*, and *p21waf1*, were markedly inhibited in Irf8-deficient BMDMs following HU or MMC stimulation (Fig. S6b). Importantly, reconstitution of *Irf8⁻/⁻* BMDMs with wild-type Irf8 but not Irf8^S151A rescued MMC-induced transcription of SASP genes (Fig. S6c). These results suggest that Irf8 is required for damage-induced cellular senescence. Dysregulated cellular senescence is frequently linked to tumorigenesis. Analysis of the Cancer Genome Atlas (TCGA) revealed that high IRF8 and STING levels were significantly correlated with the beneficial prognosis of cancer patients, such as lung adenocarcinoma, liver cancer and sarcoma (Fig. S6d). These results suggest that IRF8 and STING play potential roles in anti-tumor immune responses.

### IRF8 is abnormally activated in autoimmune syndromes

An increasing number of studies have demonstrated that STING-mediated signaling is involved in autoimmune and autoinflammatory syndromes. Recent studies have revealed that three point mutations (V147L, N154S and V155M) within STING cause SAVI[21,23,57]. Interestingly, two of the STING mutants, N154S and V155M, bound to IRF8 better than wild-type STING (Fig. 7a). Analysis of gene-profiling data indicated that IRF8 was significantly upregulated in cells and tissue samples from autoimmune and inflammatory syndrome patients comparing to healthy donors (Fig. 7b). Interestingly, in peripheral blood mononuclear cell (PBMC) derived from SLE patients, phosphorylation of IRF8^S151 and STING^S366 were markedly increased, indicating that elevated phosphorylation of IRF8^S151 is correlated with hyperactivation of STING in SLE (Fig. 7c). Considering that cGas-Sting axis mediates autoimmune phenotypes in *Trex1⁻/⁻* mice, we investigated the roles of Irf8 in autoimmunity in Trex1-deficient mice. Ablation of Irf8 by CRISPR/Cas9 in *Trex1⁻/⁻* BMDCs or BMDMs inhibited upregulation of antiviral genes, including *Ifnb1*, *Cxcl10*, *Il6* and *Il1b* in these cells (Fig. 7d). These data suggest that Irf8 plays an important role in autoimmune responses triggered by Trex1-deficiency.

## Discussion

cGAS is an important innate immune sensor in most cells in response to infection of DNA pathogens and aberrant located cellular DNA. Recently, the cGAS-STING pathways have also been reported to be involved in cellular senescence[16,17], anti-tumor immunity[18,19] and various autoimmune and autoinflammatory syndromes[20–24,57]. In this

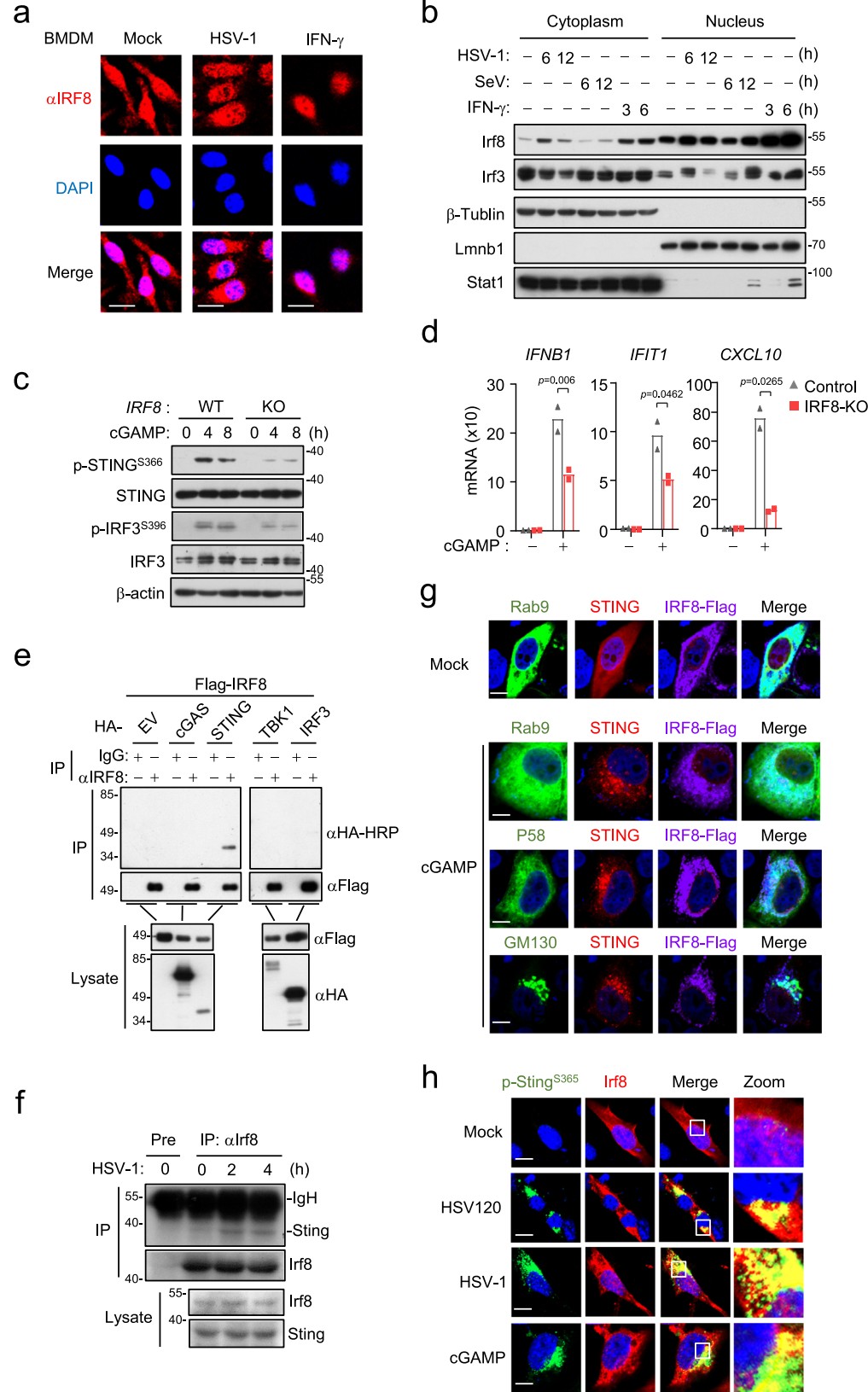

study, we reveal a transcription-independent role of IRF8 in innate immune responses to DNA by its regulation of STING-mediated signaling in monocytic cells.

Several evidences suggest that Irf8 regulates Sting activity independent of its transcriptional role. Firstly, Irf8-deficiency impaired HSV-1-induced phosphorylation of Sting[S365] and Irf3[S388] but not SeV-induced phosphorylation of Tbk1[S172] and Irf3[S388] in BMDMs and BMDCs. It is well known that phosphorylation of Sting[S365] and Irf3[S388] are immediate downstream events of cGAMP-binding to Sting after DNA virus infection and which is independent of transcription. Second, Irf8-deficiency also impaired HSV-1-induced translocation of Irf3 to the nucleus. Third, HSV-1-induced transcription of *Ifnb1* and *Cxcl10* genes

**Fig. 3 | Irf8 is association with Sting following DNA stimulation.**
**a** Immunostaining of Irf8 (red) in BMDMs un-stimulated or stimulated with HSV-1 or IFN-γ (100 ng/ml) for 6 h. Scale bars, 50 μm. **b** Cell fractionation analysis of BMDMs infected with HSV-1 or SeV, or treated with IFN-γ (100 ng/mL) for the indicated times. The nuclear and cytoplasmic extracts were analyzed by immunoblotting with the indicated antibodies. **c** Immunoblot analysis of the indicated proteins in IRF8-KO or control THP-1 cells treated with 2'3'-cGAMP for the indicated times. **d** qPCR analysis of *IFNB1*, *IFIT1* and *CXCL10* mRNA in IRF8-KO or control THP-1 cells treated with 2'3'-cGAMP for 4 h. Data are presented as mean ± AD of one representative experiment, which was repeated for 2 times with similar results. *n* = 2 technical repeats. Data were analyzed by unpaired two-tailed Student's t-test. **e** IRF8 interacts with STING in mammalian overexpression system. HEK293T cells were transfected with the indicated plasmids for 24 h. Co-immunoprecipitation and immunoblotting

analysis were performed with the indicated antibodies. EV, empty vector.
**f** Endogenous association of Irf8 and Sting in BMDM cells. The cells were left uninfected or infected with HSV-1 for the indicated times. Co-immunoprecipitation and immunoblotting analysis were performed with the indicated antibodies.
**g** Immunostaining analysis of IRF8 localization in HeLa transfected with IRF8-Flag (violet), STING-cherry (red) and the indicated GFP-tagged marker plasmids (green) for 24 h and then un-stimulated or stimulated with 2'3'-cGAMP for 4 h. GFP- Rab9 (ER marker), GFP-p58 (ERGIC marker), and GFP-GM130 (Golgi marker). Scale bars, 10 μm. **h** Immunostaining of p-Sting^S365 (green) and Irf8 (red) in murine lung fibroblasts stably transduced with Irf8 and infected with HSV-1 for 6 h or transfected with 2'3'-cGAMP and HSV120 for 4 h. Scale bars, 20 μm. Source data are provided as a Source data file.

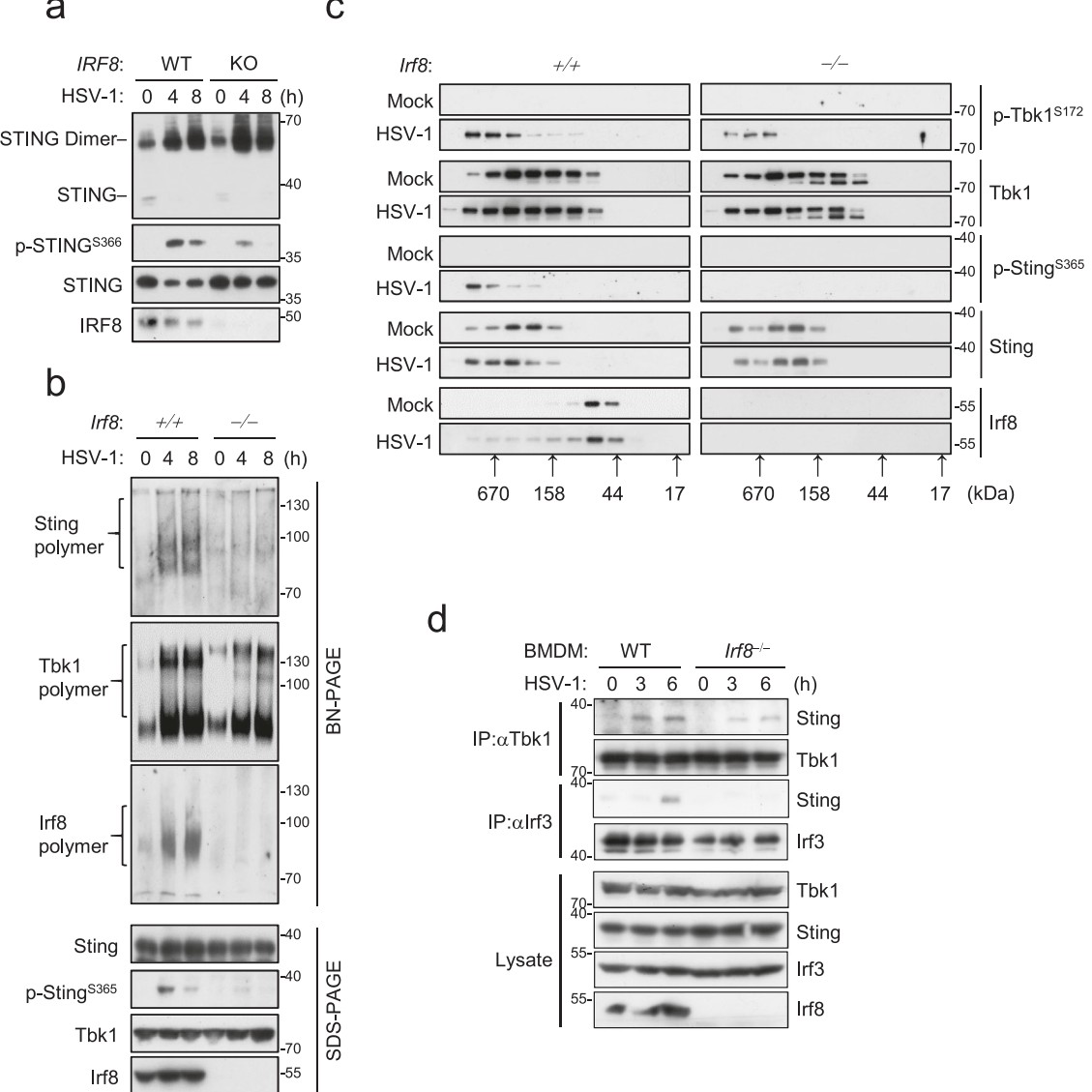

**Fig. 4 | Irf8 promotes Sting polymerization and its recruitment of Irf3.**
**a** Analysis of STING dimerization in WT and IRF8-KO THP-1 cells infected with HSV-1 for the indicated times. The lysates were fractionated by non-reducing SDS-PAGE or SDS-PAGE and analyzed with the indicated antibodies. **b** Analysis of Sting polymerization in WT and *Irf8^-/-* BMDMs infected with HSV-1 for the indicated times. The lysates were fractionated by Blue Native PAGE or SDS-PAGE and analyzed by immunoblots with the indicated antibodies. **c** Gel filtration chromatography in WT and *Irf8^-/-* BMDMs. BMDMs were left uninfected and infected with HSV-1 for 4 h

before lysis. The individual fractions were analyzed by immunoblotting with the indicated antibodies. Fraction sizes were calibrated with the gel filtration standard (Bio-Rad 151-1901). **d** Endogenous association of Sting with Tbk1 and Irf3 in WT and *Irf8^-/-* BMDMs. The cells were left uninfected or infected with HSV-1 for the indicated times. Co-immunoprecipitation and immunoblotting analysis were performed with the indicated antibodies. Source data are provided as a Source data file.

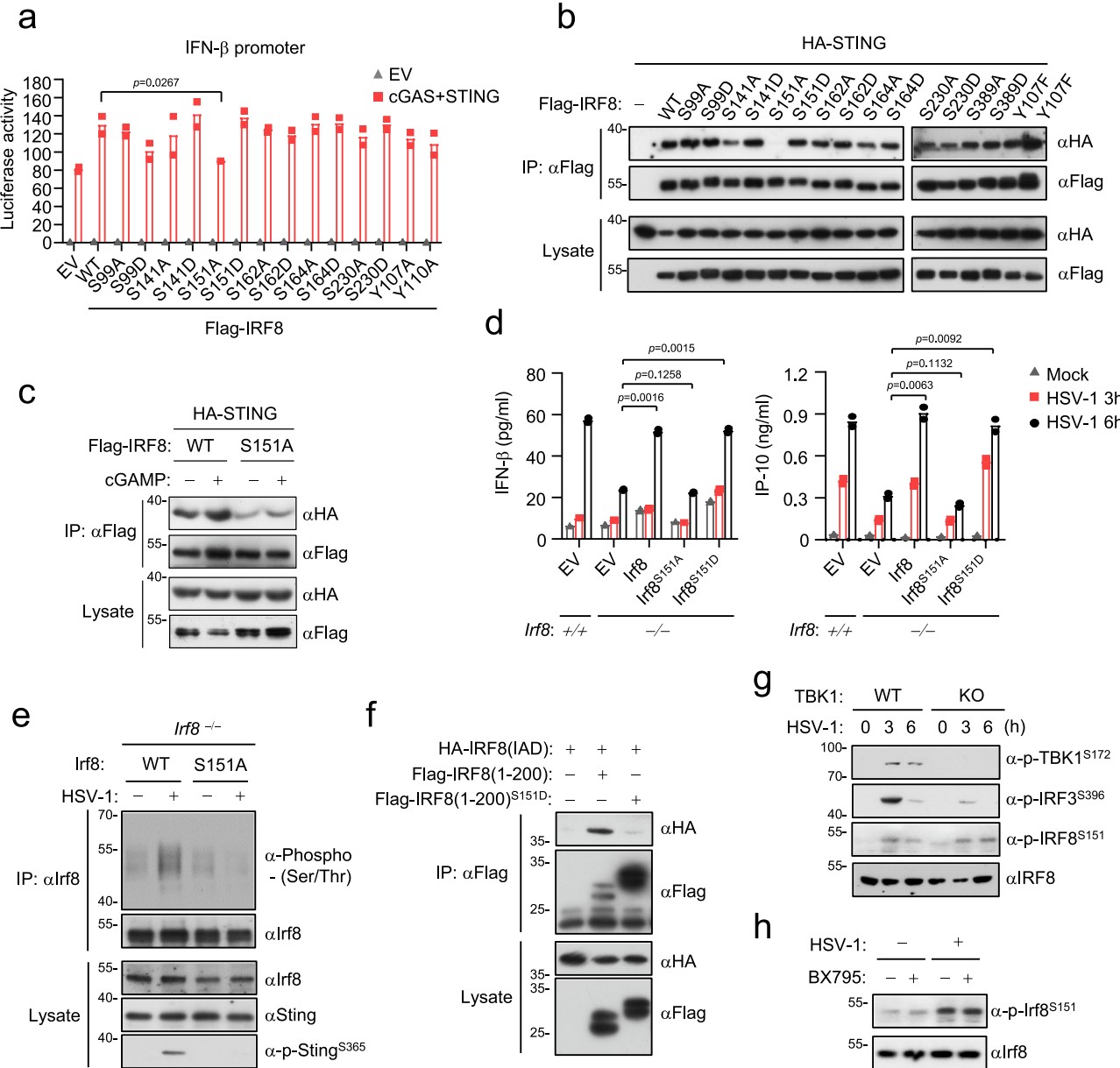

**Fig. 5 | Phosphorylation of Irf8S151 is essential for Sting-mediated signaling.** **a** Reporter assays for *IFNB1* promoter activity in HEK293T cells transfected with the indicated plasmids for 24 h. Data are presented as mean ± AD, n = 2 technical repeats. Data were analyzed by unpaired two-tailed Student's t-test. **b** Interactions of STING with IRF8 and its mutants. HEK293T cells were transfected with the indicated plasmids for 20 h followed by co-immunoprecipitation and immuno-blotting analysis with the indicated antibodies. EV, empty vector. **c** Effects of cGAMP stimulation on association of Sting with Irf8 or Irf8S151A. HEK293 cells were transfected with the indicated plasmids for 24 h and then untreated or treated with cGAMP for 4 h. Co-immunoprecipitation and immunoblotting analysis were per-formed with the indicated antibodies. **d** ELISA analysis of Ifn-β and IP-10 secretion in WT and *Irf8⁻/⁻* BMDMs reconstituted with the indicated plasmids. Cells were left uninfected or infected with HSV-1 for the indicated times. Data are presented as

mean. n = 2 technical repeats. Data were analyzed by unpaired two-tailed Student's t-test. **e** Detection of Irf8 phosphorylation in *Irf8⁻/⁻* BMDMs reconstituted with Irf8 or Irf8S151A. Cells were left uninfected or infected with HSV-1 for the indicated times followed by coimmunoprecipitation and immunoblotting analysis with the indi-cated antibodies. **f** Effects of S151 mutation on the association of its N-terminal region and IAD. The experiments were performed similarly as in **b**. **g** HSV-1-induced phosphorylation of Irf8S151, Irf3S388 and Tbk1S172 in WT and Tbk1-knockout BMDMs. Cells were left uninfected or infected with HSV-1 for the indicated times. Immu-noblotting analysis was performed with the indicated antibodies. **h** The Tbk1 inhibitor BX795 does not affect HSV-1-induced phosphorylation of Irf8S151. BMDMs were untreated or treated with BX795 (1 µM) and then infected with HSV-1 for the indicated times. Immunoblotting analysis was performed with the indicated anti-bodies. Source data are provided as a Source data file.

and secretion of Ifn-β and IP-10 in *Irf8⁻/⁻* BMDMs were fully rescued by reconstitution with WT and Irf8 mutants that lack transcriptional activity. Most importantly, the Irf8 truncation mutant in which the N-terminal DNA binding domain is deleted, fully rescued HSV-1-induced expression of downstream antiviral genes in Irf8-deficient BMDMs.

Our results suggest that Irf8 acts as a critical component linking Sting-mediated Irf3 activation after DNA virus infection. Confocal and

biochemical experiments indicated that Irf8 was barely associated with Sting in un-infected cells, and their association was markedly increased early after HSV-1 infection. Irf8-deficiency impaired HSV-1-induced recruitment of Irf3 to Sting, but had no dramatic effects on the recruitment of Tbk1 to Sting. Our results also indicated that Irf8-deficiency impaired HSV-1-induced phosphorylation of StingS365 and Irf3S388 but not Tbk1S172. Consistently, Irf8-deficiency impaired the polymerization of Sting, an important step for its recruitment and

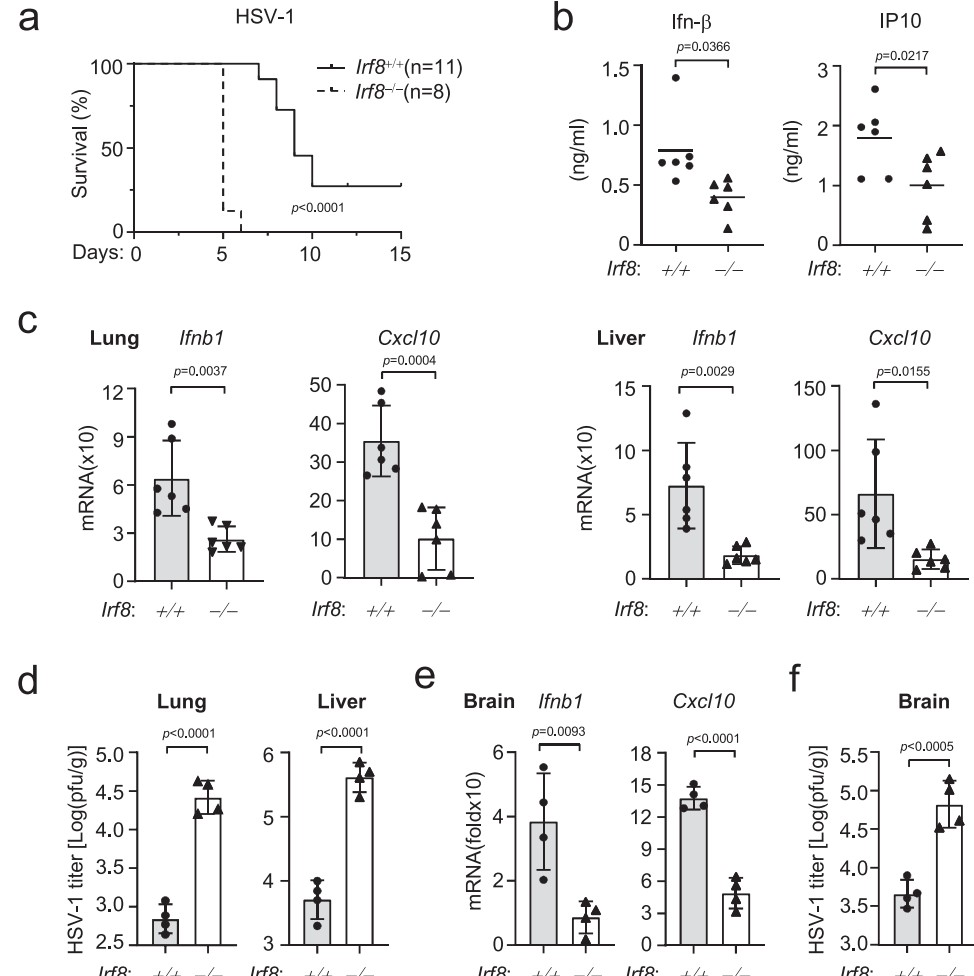

**Fig. 6 | Irf8 is important for innate immunity to DNA virus in vivo. a** Survival kinetics of WT (*n* = 11, female, 8 weeks old) and *Irf8⁻/⁻* mice (*n* = 8, female, 8 weeks old) after injection with HSV-1 (1 × 10⁷ PFU per mouse). Data was analyzed by log-rank test. **b** Serum cytokine levels of WT and *Irf8⁻/⁻* mice (*n* = 6 per strain, female, 8 weeks old) injected intraperitoneally with HSV-1 (1 × 10⁷ PFU per mouse) for 6 h. **c** qPCR analysis of *Ifnb1* and *Cxcl10* mRNA levels in lungs (left) and livers (right) of WT and *Irf8⁻/⁻* mice (*n* = 6 per strain, female, 8 weeks old) injected intraperitoneally with HSV-1 (1 × 10⁷ PFU per mouse) for 24 h. **d** Plaque assays for viral titers in lungs and livers of WT and *Irf8⁻/⁻* mice (*n* = 4 per strain, female, 8 weeks old) injected intraperitoneally with HSV-1 (1 × 10⁷ PFU per mouse) for 3 days. **e, f** qPCR analysis of *Ifnb1* and *Cxcl10* mRNA levels (**e**) and plaque assays for HSV-1 titers (**f**) in brains of WT and *Irf8⁻/⁻* mice (*n* = 4 per strain, female, 8 weeks old) injected intraperitoneally with HSV-1 (1 × 10⁷ PFU per mouse) for 4 days. Data in **b**–**f** are shown as mean ± SEM. Data were analyzed by unpaired two-tailed Student's t-test. Source data are provided as a Source data file.

activation of Irf3. We also showed that overexpression of Irf8 relieved the auto-inhibition of Sting. In light of these results and previous studies on the mechanisms of Sting-mediated Irf3 activation, we propose a working model on how Irf8 modulates innate immune response to DNA virus. Upon DNA virus infection, binding of cGas to viral DNA induces the synthesis of cGAMP, which binds to the ER-located Sting and promotes its dimerization. The dimerization of Sting causes its conformational changes, which facilitates the recruitment of Tbk1 and Irf8. In this complex, Tbk1 is autophosphorylated independently of Irf8, whereas Irf8 may relieve the auto-inhibition of Sting and promotes its polymerization. The polymerization of Sting conditions it for Tbk1-mediated phosphorylation at S365, which in turn promotes the recruitment of Irf3 to Sting polymers. The recruited Irf3 is then phosphorylated by Tbk1 at S388, leading to its activation and induction of downstream antiviral genes. It has been previously shown that phosphorylation of Sting^S365 by Tbk1 is required for its recruitment and activation of Irf3 and does not affect Sting-mediated NF-κB activation[14,58]. Consistently, our results indicated that Irf8-deficiency impaired Sting-mediated Irf3 activation but not NF-κB activation or autophagy. In fact, it is interesting that HSV-1 or VCAV-triggered NF-κB signaling was even increased in Irf8-deficient cells. The simplest

explanation is that Irf8 acts as a modifier or balancer in monocytes by facilitating recruitment of Irf3 to S365-phosphorylated Sting while weakly suppressing the recruitment of the IKK complex to Sting, therefore, is differentially required for Sting-mediated effects.

Our experiments also indicate that phosphorylation of Irf8^S151 is essential for HSV-1-induced phosphorylation and activation of Sting. It is noted that the N-terminus of Irf8 can interact with its IAD, but the N-terminus of Irf8 with S151D mutation losses its ability to interact with the IAD. These results suggest that phosphorylation of S151 of Irf8 may release it from an auto-inhibitory status, which promotes the polymerization of Sting. Consistently, mutation of S151 of Irf8 to alanine abolishes its ability to mediate HSV-1-induced expression of downstream antiviral genes. Since the phosphorylation of Irf8 S151 following HSV-1 infection is not affected in Tbk1-deficient cells, a kinase other than Tbk1 is responsible for this phosphorylation. Our results also showed that Irf8^S151 was basally phosphorylated and which was increased following HSV-1 infection. It is unknown whether phosphorylation of Irf8^S151 occurs before or after its recruitment to Sting. These remaining questions need to be investigated in future studies.

Unlike STING, which is widely expressed in most types of cells, IRF8 is mainly expressed in lymphoid and myeloid cells. This raises an

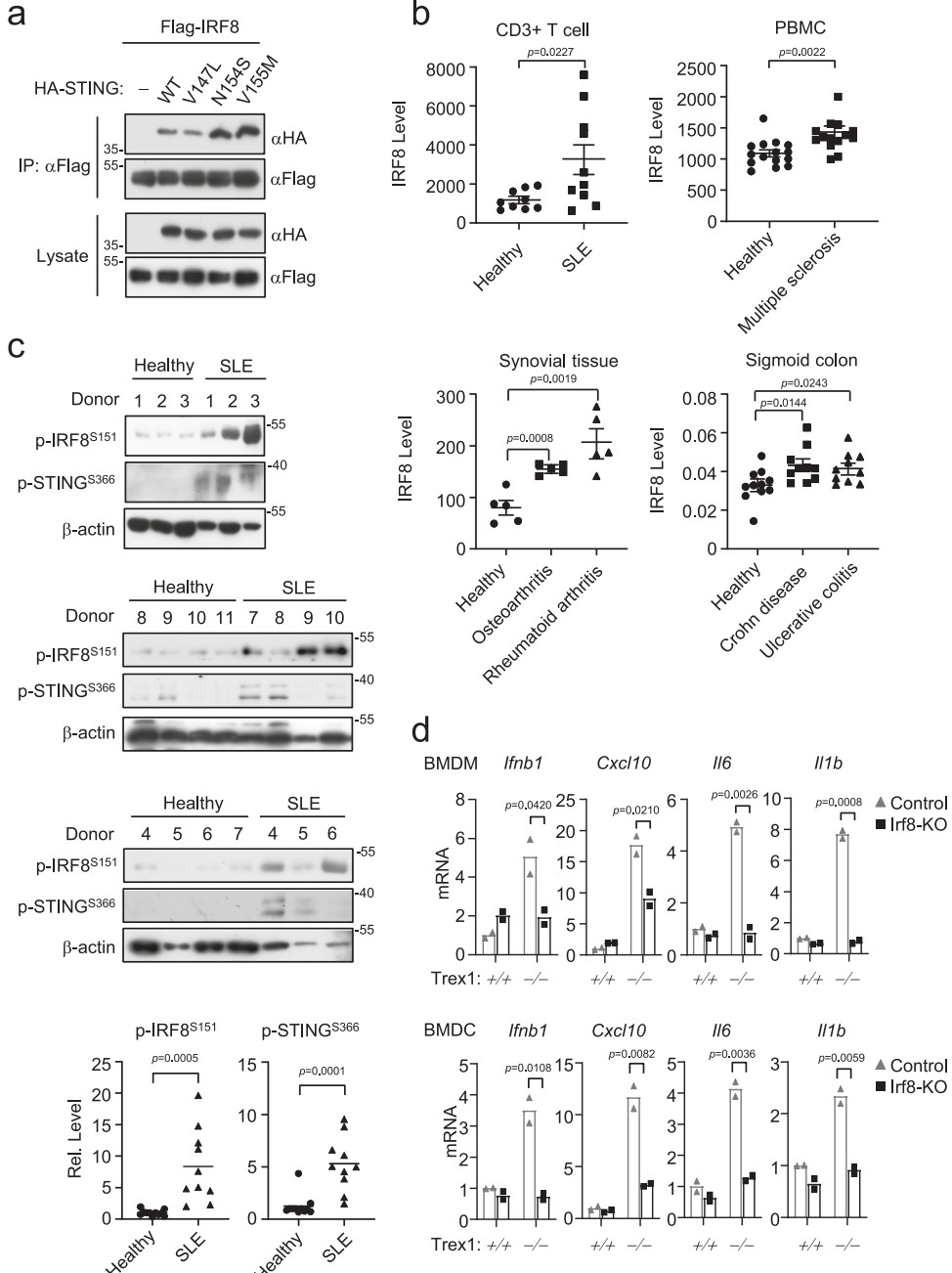

**Fig. 7 | IRF8 is abnormally activated in autoimmune syndromes.**
**a** Coimmunoprecipitation of IRF8 and STING SAVI mutants in HEK293T cells. Cells were transfected with the indicated plasmids for 20 h followed by coimmunoprecipitation and immunoblotting analysis with the indicated antibodies. **b** Profiling analysis of IRF8 expression in healthy donors and patients with autoimmune diseases. Graph shows mean ± SEM, $n$ = 5–15 independent samples. Data were analyzed by unpaired two-tailed Student's t-test. **c** Immunoblotting analysis of IRF8$^{S151}$ and STING$^{S366}$ in PBMCs from healthy donors ($n$ = 11 independent samples) and SLE patients ($n$ = 10 independent samples). Relative phosphorylation abundance of IRF8$^{S151}$ and STING$^{S366}$ relative to β-actin was quantitated. Data were analyzed by unpaired two-tailed Student's $t$ test. **d** qPCR analysis of *Ifnb1*, *Cxcl10*, *Il6* and *Il1b* mRNA levels in WT or *Trex1*$^{-/-}$ BMDMs and BMDCs with or without additional Irf8 knockout by CRISP-Cas9. Data are presented as mean, $n$ = 2 technical repeats. Data were analyzed by unpaired two-tailed Student's $t$ test. Source data are provided as a Source data file.

interesting question why regulation of STING-mediated innate immune responses does not require IRF8 in cells that do not express IRF8. It is possible that IRF8 is not required or another protein acts in a similar way for STING-mediated innate immune responses in other cell types. A unique role of Irf8 in monocytes also provides an explanation on the notion that these cells express higher levels of antiviral genes following viral infection than other cells.

Our results indicated that Irf8-deficiency inhibits DNA but not RNA virus-induced transcription of downstream antiviral genes in monocytes and well-differentiated macrophages and DCs. Irf8-deficiency inhibits HSV-1-induced production of serum IFN-β and IP-10, increases HSV-1 replication in the tissues, and makes the mice more susceptible to HSV-1-induced death. In this context, it has been previously shown that *Irf8*$^{-/-}$ mice are also more susceptible to the DNA viruses such as VACV and MCMV, but have an intact antiviral response to the RNA virus VSV[43,44]. Since Irf8 also plays important roles in other immunological processes, the Irf8-Sting axis may not be fully responsible for the observations with our in vivo studies. However, these

results together with our cellular and biochemical experiments that clearly demonstrated a role of Irf8 in Sting- but not Visa/Mavs-mediated innate immune response in monocytes, suggest that IRF8 plays a direct role in innate immune responses to DNA viruses rather than a function secondary to the effects of differentiation of monocytes.

Consistent with a role of Irf8 in Sting-mediated innate immune responses in mice, analysis of gene-profiling data shows that IRF8 is significantly upregulated in cellular and tissue samples from patients with autoimmune or inflammatory syndromes. Elevated activation of IRF8 and STING are also detected in PBMCs from SLE patients. Moreover, Irf8-deficiency inhibits autoimmune phenotypes in BMDCs and BMDMs derived from *Trex1*[-/-] mice. These observations suggest that heightened activation of IRF8 and STING is associated with autoimmune diseases.

In addition to antiviral immunity, cGAS-STING axis is also involved in cellular senescence and anti-tumor immunity. Our results show that IRF8 is important for damage-induced cellular senescence in macrophages and this function requires its interaction with STING. Abnormity of cellular senescence is frequently linked to tumorigenesis. Analysis of TCGA indicates that high STING and IRF8 expression levels are significantly correlated with the beneficial prognosis of cancer patients, such as lung adenocarcinoma, liver cancer and sarcoma. These findings suggest that IRF8, like STING, also plays an important role in the prevention of tumorigenesis. In conclusion, the findings provided in this study point to a direct role of IRF8 in STING-mediated innate immune responses, which contributes to our understanding of the complicated mechanisms of autoimmune diseases and tumorigenesis.

## Methods

### Reagents, antibodies, viruses and cells
LMW-Poly(I:C), LPS, 2′3′-cGAMP and DMXAA (InvivoGen); hydroxyurea, camptothecin and mitomycin C (MCE); GM-CSF, Flt3L (peproTech); lipofectamine 2000 (Invitrogen); polybrene (Millipore); RNAiso Plus (Takara); HT-DNA (Sigma); SYBR (BIO-RAD); dual-specific luciferase assay kit (Promega); ELISA kit for murine Ifn-β (PBL); ELISA kits for murine IP-10 (Biolegend) were purchased from the indicated manufacturers. DNA oligonucleotides HSV120, VACV70, DNA90 and ISD45 were synthesized by Sangon Biotech, and their sequences is shown in Supplementary Table 1.

Information on the commercially available antibodies used in this study is provided in Supplementary Table 2. Anti-STING and anti-IRF8 polyclonal antibodies were generated by immunizing rabbits or mice with purified STING(151–379) and IRF8 proteins. Antisera against phosphor-IRF8^S151 were generated by immunizing rabbits with the synthetic peptide of mouse IRF8 ($_{145}$ELIKEPS$^P$VDE$_{154}$) by ABclonal Technology (Wuhan).

SeV, VSV, NDV, HSV-1, and VACV have been previously described[15].

HEK293, Vero and THP-1 cells were obtained from ATCC. HEK293T cells were provided by Dr. Hong-Bing Shu (Wuhan University).

### Constructs
IFN-β and ISRE luciferase reporter plasmids, as well as mammalian expression plasmids for HA- or Flag-tagged STING and its mutants, cGAS, TBK1, IRF3, and IRF3-5D have been previously described[15]. pcDNA3.1-Flag-cGAS was provided by Dr. Zhijian James Chen (University of Texas Southwestern Medical Center). HA-, Flag- and GFP-tagged IRF8 and its truncations were constructed by standard molecular biology techniques.

### Cell culture
Vero, HEK293T and HEK293 cells were grown in complete DMEM (Hyclone) supplemented with 10% ($v/v$) fetal bovine serum (Hyclone), 2 mM L-glutamine, 10 mM HEPES, and 1% ($v/v$) Penicillin/Streptomycin

(Hyclone); THP1 cells were grown in RPMI (Hyclone) containing 10% fetal bovine serum (Hyclone). All cell lines were cultured at 37 °C and 5% $CO_2$.

### Transfection and reporter assays
HEK293T and HEK293 cells were transfected by standard calcium phosphate precipitation method. THP-1 was transfected by lipofectamine 2000. Hela cells were transfected by FuGENE® HD Transfection Reagent. To normalize transfection efficiency, 0.01 μg of pRL-TK (Renilla luciferase) reporter plasmid was added to each transfection. Luciferase assays were performed using a dual-specific luciferase assay kit. For qPCR analysis and ELISA experiments, synthetic DNA (2 μg/10^6 cells), synthetic RNA (2 μg/10^6 cells) or 2′3′-cGAMP (0.2 μg/10^6 cells) was transfected into cells for 3 h by Lipofectamine 2000.

### Mice and genotyping
*Irf8*[-/-] mice on a C57BL/6 background were purchased from Center for Animal Experiment/Animal Biosafety Level-III Laboratory, Wuhan University, China. The genotypes of the wild-type and *Irf8*[-/-] mice were confirmed by sequencing PCR products amplified from the genomic DNAs isolated from mouse tails using the following primers: #1: 5′-CATGGCACTGGTCCAGATGTCTTCC-3′, #2: 5′-CTTCCAGGGGA-TACGGAACATGGTC-3′ and #3: 5′-CGAAGGAGCAAAGCTGC-TATTGGCC-3′. Amplification of the wild-type allele with primer #1 and #2 results in a 258-bp fragment, whereas amplification of the knockout allele with primer #1 and #3 results in a 548-bp fragment.

*Trex1*[-/-] mice on a C57BL/6 background were kindly provided by Dr. Bo Zhong (Wuhan University, Wuhan). *Irf3*[-/-] mice on a C57BL/6 background were kindly provided by Dr. Xin-Wen Chen (Wuhan Institute of Virology, Wuhan). The production strategies and genotyping of these mice have been previously described[59].

All mice were housed in groups of 5 mice per cage on a 12 h light/dark cycle in a temperature-controlled specific pathogen-free (SPF) room (23–25 °C) and relative humidity of 40–70% with free access to water and food. At the experimental endpoint, animals were sacrificed by cervical dislocation after isoflurane anesthesia. Animal experiments were conducted without blinding, with 6–8 week old age- and sex-matched mice. All animal experiments were performed in accordance with the Guideline for Animal Care and Use of Wuhan Institute of Virology, Chinese Academy of Sciences.

### Isolation of PBMCs from human blood samples
The blood samples of SLE patients and healthy people were collected by Drs. Fu-Bing Wang (Zhongnan Hospital of Wuhan University, China). The use of samples in this study were approved by the Medical Ethics Committee of Zhongnan Hospital of Wuhan University (No. 2019022). PBMCs were isolated with SepMate™ (86415, STEMCELL) according to the manufacturer's instructions.

### Preparations of bone marrow-derived macrophages and DCs
Mouse bone marrow cells were isolated from tibia and femur. For preparations of BMDMs, the bone marrow cells were cultured in 10% M-CSF-containing conditional medium from L929 cells for 3–5 days. For preparations of BMDCs, the bone marrow cells were cultured in medium containing murine GM-CSF (50 ng/ml) for 6–9 days.

To differentiate bone marrow-derived cDCs, bone marrow cells were suspended in Iscove's Modified Dulbecco's Medium (IMDM) supplemented with 10% FBS, 1% Penicillin Streptomycin solution, 1% Sodium Pyruvate, 1% MEM non-essential amino acid, 1% L-glutamine solution, and 55 mM β-mercaptoethanol (complete IMDM) and were cultured with Flt3L (25 ng/ml) conditioned medium for 7 to 8 days. The cells were then stained with fluorescent antibodies before sorting. pDCs were Bst2^- B220^+, cDC1s were Bst2^- B220^- CD11c^+ MHCII^+ CD24^+ CD172a^-, cDC2s were Bst2^- B220^- CD11c^+ MHCII^+ CD172a^+. The antibodies used in FACS were APC anti-mouse CD317 (BST2, PDCA-1)

antibody (0.06 μg/10[6] cells, Biolegend), PE anti-mouse/human CD45R/B220 antibody (0.25 μg/10[6] cells, Biolegend), APC/Cyanine7 anti-mouse CD11c antibody (1 μg/10[6] cells, Biolegend), PE/Cyanine7 anti-mouse CD24 antibody (0.5 μg/10[6] cells, Biolegend), PerCP/Cyanine5.5 anti-mouse CD172a (SIRPα) antibody (1.5 μg/10[6] cells, Biolegend) and Pacific Blue™ anti-mouse I-A/I-E antibody (0.25 μg/10[6] cells, Biolegend).

## CRISPR-Cas9 knockout

Genome engineering was performed with the CRISPR/Cas9 system[60]. Briefly, double-stranded oligonucleotides corresponding to the target sequences were cloned into the lenti-CRISPR-V2 vector, which was co-transfected with packaging plasmids into HEK293 cells. Two days after transfection, the viruses were harvested and used to infect target cells. The infected cells were selected with puromycin (1 μg/ml) for at least 5 days. The information of gRNA sequences is shown in Supplementary Table 3.

## Cell lines and retroviral gene transfer

Transduction of IRF8-RNAi plasmid to THP-1 cells were performed by retroviral-mediated gene transfer. Briefly, HEK293 cells plated on 100 mm dishes were transfected with the indicated retroviral plasmid (10 μg) together with the pGag-pol (10 μg) and the pVSV-G (3 μg) plasmids. Two days after transfection the viruses were harvested and used to infect the indicated cells in the presence of polybrene (4 μg/ml). The infected cells were selected with puromycin (0.5–2 μg/ml) for at least 4 days.

## Quantitative PCR

Total RNA was isolated for qPCR analysis to measure mRNA abundance of the indicated genes. The relative abundance of the indicated mRNA derived from mouse or human cells was normalized to *Actb* or *GAPDH* mRNA level respectively. Gene-specific primer sequences were described in Supplementary Table 4.

## Gene set enrichment analysis (GSEA)

After data pre-processing, the enrichment analyses for cytosolic DNA-sensing pathway gene sets were analyzed with the GSEA V4.2.3 software. Gene set databases used during this analysis were downloaded from the KEGG (https://www.genome.jp/entry/mmu04623).

## ELISA

BMDMs and THP-1 cells were stimulated with viruses or transfected with synthetic nucleic acids for the indicated times. The culture media were collected for measurement of IP-10 and IFN-β by ELISA. The mouse serum was collected at 6 h after infection for measurement of cytokine production by ELISA.

## Co-immunoprecipitation and immunoblot analysis

HEK293T, THP-1 or BMDMs were lysed in l ml NP-40 lysis buffer (20 mM Tris-HCl (pH 7.4), 150 mM NaCl, 1 mM EDTA, 1% Nonidet P-40, 10 μg /ml aprotinin, 10 μg/ml leupeptin, and 1 mM phenylmethylsulfonyl fluoride). For each immunoprecipitation reaction, a 0.8 ml aliquot of lysate was incubated with 0.5–2 μg of the indicated antibody or control IgG and 35 μl of a 1:1 slurry of Protein-G Sepharose (GE Healthcare) at 4 °C for 3 h. The Sepharose beads were washed three times with 1 ml of lysis buffer containing 500 mM NaCl. The precipitates were fractionated by sodium dodecyl sulfate–polyacrylamide gel electrophoresis (SDS-PAGE), and immunoblotting analysis was performed with the indicated antibodies.

## Confocal microscopy

Confocal microscopy was performed as previously described[15]. Briefly, cells were fixed with 4% paraformaldehyde for 10 min at 25 °C and then permeabilized and stained with the indicated antibodies by standard protocols. The stained cells were observed with an Olympus confocal microscope under a 60× oil objective.

## Gel filtration chromatography

BMDMs cells were cooled and centrifuged at $300 \times g$ for 5 min at 4 °C. The cells ($1.5 \times 10^8$ cells per sample) were lysed in lysis buffer (20 mM Tris–HCl, pH 7.4, 150 mM NaCl, 1 mM EDTA, 1% NP-40, supplemented with protease and phosphatase inhibitors) and sonicated for 1 min. Lysates were centrifuged at $13,000 \times g$ for 30 min at 4 °C, followed by filtering through a 0.45 μm syringe filter to clarify the lysates. The cleared lysates were subjected to gel filtration chromatography using Superdex 200 Increase 10/300 GL column and separation buffer (50 mM Tris, pH 7.4, 150 mM NaCl). Fraction collector collected 1 ml per fraction for 12 fractions after the 0.2 CV dead volume. A gel filtration standard (Bio-Rad 151-1901) was also run to calibrate the fractions.

## STING dimerization assay

STING dimerization assays by non-reducing SDS-PAGE were performed as previously described[61]. In brief, BMDM and THP-1 cells were lysed on ice for 15 min in lysis buffer (10 mM PIPES-KOH [pH 7.0], 50 mM NaCl, 5 mM MgCl2, 5 mM EGTA, 1% NP40, 10% glycerol, and a mixture of protease inhibitors and phosphatase inhibitors). The lysates were mixed with 5×SDS sample buffer without 2-ME. Samples were separated by electrophoresis on 10% SDS-PAGE.

## Digitonin permeabilization

cGAMP was delivered to cells pretreated with digitonin permeabilization solution (50 mM HEPES [pH 7.0], 100 mM KCl, 3 mM MgCl2, 0.1 mM DTT, 85 mM Sucrose, 0.2% BSA, 1 mM ATP, 0.1 mM GTP and 10 μg/ml digitonin) at 37 °C for 30 min.

## Blue Native-PAGE

Cells were lysed in native lysis buffer (10% glycerol, 25 mM NaCl, 20 mM HEPES pH 7.0, 1% DDM, and complete protease inhibitors) and then solubilized by rotating 30 min at 4 °C. Lysates were centrifuged for 10 min at 14000 rpm. The supernatant was added with 4× native sample buffer (Invitrogen), run on a native PAGE gel, and transferred to a PVDF membrane using a wet transfer system (BioRad) before immunoblotting analysis with the indicated antibodies.

## Viral infection in mice

Age- and sex-matched wild-type and *Irf8*[−/−] mice were injected intraperitoneally with HSV-1. The viability of the infected mice was monitored for 15 days. Tissues from infected mice were weighed and homogenized with PBS, followed by centrifugation at $1620 \times g$ for 30 min. The supernatants were collected for plaque assays on monolayers of Vero cells seeded in 24-well plates.

## Cellular senescence assays

To evaluate damage-induced cellular senescence, cells at approximately 60-70% confluence were treated with hydroxyurea (10 mM), CPT (1 μM) or mitomycin C (1 μM) for 24 h and then changed to fresh medium. The cells were cultured for another 5 d and harvested for the SA-β-Gal assay using a cellular senescence assay kit (abbkine) according to the manufacturer's manual. For quantification of β-galactosidase staining positive cells, the blue positive cells in at least three randomly selected fields were counted under a microscope. To detect the SASPs, cells were lysed after 5–8 d of stimulation and subjected to RNA extraction and qPCR assays to detect the mRNA levels of MMP12, IL6, and p21waf1.

## Statistics and reproducibility

Unpaired Student's *t*-test was used for statistical analysis with Microsoft Excel and GraphPad Prism Software. For the mouse survival study,

Kaplan–Meier survival curves were generated and analyzed by Log-Rank test; $P < 0.05$ was considered significant. Fluorescent-imaging analysis was performed in a blinded fashion. Densitometry quantification was made by ImageJ Software. All data are representative of at least two independent experiments with similar results.

### Reporting summary

Further information on research design is available in the Nature Research Reporting Summary linked to this article.

## Data availability

All the data supporting the findings of this study are available within the article and its supplementary information files, or can be obtained from the corresponding author upon reasonable request. A reporting summary for this article is available as a Supplementary Information file. Source data are provided with this paper.

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

## Acknowledgements

This study was supported by grants from the National Natural Science Foundation of China (31870869, 32022026, W.-W.L.), the Strategic Priority Research Program (XDB29010302, Y.-Y.W.), the Key Research Programs of Frontier Sciences funded by Chinese Academy of Sciences (ZDBS-LY-SM033, W.-W.L.) and the Youth Innovation Promotion Association CAS (No. 2019327, W.-W.L.). We thank the Center for Experiment Animal of Wuhan Institute of Virology for animal experiments. We thank the Institutional Center for Shared Technologies and Facilities of Wuhan Institute of Virology for technical support. We thank members of the Wang lab for helpful discussions.

## Author contributions

W.-W.L. and Y.-Y.W. conceived and designed the study. W.-W.L., Z.T., P.C., Z.-Q.Z., and S.-Y.W. performed the majority of experiments. Y.L. performed the flow cytometry and cell sorting experiments. F.-B.W. provided human blood samples in this study. W.-W.L., Z.T., P.C., Z.-Q.Z., S.-Y.W., Y.L., S.L., and Y.-Y.W. analyzed the data. W.-W.L. and Y.-Y.W. wrote the manuscript. All of the authors discussed the results and commented on the manuscript.

## Competing interests

The authors declare no competing interests.
