## [Peer Review File · Nature Communications]

Transcription-independent regulation of STING activation and innate immune responses by IRF8 in monocytesREVIEWER COMMENTS

Reviewer #1 (Remarks to the Author):

In this work, Luo et al report a role for IRF8 in STING signaling. The authors show that IRF8-deficiency renders cells and mice unable to mount IFN responses to cGAMP, dsDNA, and HSV1, and propose that this leads to impaired host defense in vivo. At the mechanistic level, the authors show that IRF8 supports STING oligomerization and IRF3 activation, but does not impact on TBK1 phosphorylation. This suggests a role for IRF8 at the crucial step of IRF3 recruitment/activation in the STING signaling pathway. The work is very interesting, and provides information on a novel factor in STING signaling. But the manuscript could be improved further by addition of more in-depth information on mechanistic details on IRF8

1. Is IRF3 recruited to STING in IRF8-deficient cells? Can be tested by co-IP.
2. Fig.2: Please double-check whether the pIRF3 blot is for S388 as said in panel a.
3. The observation that TBK1 phosphorylation is unchanged, would that TBK1 recruitment to STING is not impaired in IRF8-deficient cells. This should be tested by co-IP.
4. The data presented could suggest that IRF8 facilitates recruitment of IRF3 to STING.
5. Figure 3f. This data suggests that IRF8 is recruited to STING after HSV1 infection, but with a rather delayed kinetics (6-9 h). This does not explain the early effect on HSV1-induced IFN β expression, 2h, Fig 1a). The authors need to expand on this, including new data.
6. Along the same lines, does the IRF8 S151A mutant retain the ability to interact with STING in a manner inducible by cGAMP treatment?
7. Furthermore, the early effect of IRF8 KO on STING-induced IFN/ISG expression could potentially be explained by alteration of constitutive gene expression. RNAseq of unstimulated control versus KO cells could reveal whether IRF8 KO causes a more global alteration of genes related to STING signaling.
8. Where in the cell does IRF8 associate with STING? BFA treatment combined with co-IP or ImageStream analysis could be performed.
9. Figure 6a. The in vivo data may overestimate the effect of the described IRF8-STING mechanism on host defense, since IRF8 is involved in many other immunological processes of relevance for antiviral defense (as nicely described and cited by the authors; ref 44 reports on a role for IRF8 in control of the RNA virus LCMV). It would be nice to see whether *Irf8*^{-/-}*Sting*^{-/-} show the same phenotype as the single KO mouse strains. As a minimum, the authors should discuss this issue.

Reviewer #2 (Remarks to the Author):

In this manuscript by Luo et al, the authors identified a new function of IRF8 in regulating STING signaling. IRF8 is a transcription factor playing important roles in development of myeloid cells especially DCs, NK cell activity, inflammasome activation, all of which acting through its transcriptional activity. The authors identified a new cytoplasmic activity of IRF8 in this manuscript that promotes STING signaling complex assembly and IFN signaling.

The authors presented a thorough and convincing set of STING signaling assays clearly demonstrating that IRF8 is a positive regulator of the STING pathway. The authors pinned down the step where IRF8 acts (between TBK1 phosphorylation and STING polymerization and phosphorylation). In vivo HSV-1 infection and data on SLE and *Trex1*^{-/-} cells generally support the model.

How is IRF8 activated by DNA and how it is recruited to STING are not very clear. For example, based on published structural evidence, STING recruits TBK1 to one set of STING dimer and STING phosphorylation by TBK1 requires transacting on another set of STING dimers. The authors observed in *Irf8*^{-/-} cells pTBK1 but not pSting. Could IRF8 facilitate packing of STING dimers into polymers? The coIP studies in Fig 3E show substantial interaction at 9 hpi, which is pretty late for IRF8 to modulate STING activation. The authors should propose a reasonable model based on current understanding of the STING pathway, hopefully explain how/when DNA activates IRF8 phosphorylation, when does IRF8 interact with STING, how does that interaction impact TBK1 recruitment, STING polymerization and STING phosphorylation.

The authors showed increased NFκB signaling (p-p65, p-Iκb) and decreased IFN signaling (pSTING and pIRF3) in *IRF8*^{-/-} cells when the STING pathway is activated by HSV-1 or VACV. If STING is not activated, how do they get NFκB signaling? Both IFN and NFκB are downstream signaling pathways that require STING activation, it is possible that IRF8 controls the balance between these two downstream pathways? They should activate STING with an agonist and measure IFN, NFκB and LC3/autophagy to see if *IRF8*^{-/-} cells truly lost all STING activities.

Response Letter

Dear Editor and reviewers,

We would like to thank you for your interest in our study and very constructive comments/suggestions on our manuscript, which have helped us to improve the manuscript. We have now performed additional experiments and added certain discussions to address your concerns. Following is a point-to-point response to your concerns.

Reviewer #1:

In this work, Luo et al report a role for IRF8 in STING signaling. The authors show that IRF8-deficiency renders cells and mice unable to mount IFN responses to cGAMP, dsDNA, and HSV1, and propose that this leads to impaired host defense in vivo. At the mechanistic level, the authors show that IRF8 supports STING oligomerization and IRF3 activation, but does not impact on TBK1 phosphorylation. This suggests a role for IRF8 at the crucial step of IRF3 recruitment/activation in the STING signaling pathway. The work is very interesting, and provides information on a novel factor in STING signaling. But the manuscript could be improved further by addition of more in-depth information on mechanistic details on IRF8

1. Is IRF3 recruited to STING in IRF8-deficient cells? Can be tested by co-IP.

Reply: We have now performed the co-IP experiments as the reviewer suggested. The result showed that HSV-1-triggered recruitment of Irf3 to Sting was impaired in Irf8-deficient cells (**new Fig. 4d, page 10**).

2. Fig.2: Please double-check whether the pIRF3 blot is for S388 as said in panel a.

Reply: The S388 residue of mouse Irf3 corresponds to S396 residue of human IRF3.

In the revised manuscript, we have labeled phosphorylation of mouse Irf3 as p-Irf3^{S388}, whereas human IRF3 phosphorylation at S396 was labeled as p-IRF3^{S396}. In addition, in the revised manuscript, all letters in the name of a human protein are capitalized, while only the first letter in the name of a mouse protein is capitalized.

3. *The observation that TBK1 phosphorylation is unchanged, would that TBK1 recruitment to STING is not impaired in IRF8-deficient cells. This should be tested by co-IP.*

Reply: We have now performed the co-IP experiments as the reviewer suggested. The results showed that the recruitment of Tbk1 to Sting following HSV-1 infection was minimally affected in Irf8-deficient cells (**new Fig S4d, page 10**).

4. *The data presented could suggest that IRF8 facilitates recruitment of IRF3 to STING.*

Reply: Yes, our original and new results support the conclusion that IRF8 facilitates recruitment of IRF3 to STING.

5. *Figure 3f. This data suggests that IRF8 is recruited to STING after HSV1 infection, but with a rather delayed kinetics (6-9 h). This does not explain the early effect on HSV1-induced IFNB expression, 2h, Fig 1a). The authors need to expand on this, including new data.*

Reply: The association between Irf8 and Sting is dynamic. We have now examined endogenous interaction between Irf8 and Sting in BMDMs. As shown in **new Fig 3f (page 9)**, Irf8 was barely associated with Sting in un-infected cells, and their association was markedly increased at 2 hours after HSV-1 infection. In addition, gel filtration experiments showed that Irf8 and Sting were in the same complex in cells infected with HSV-1 for 4 hours (**Fig. 4c and S3b**). These results suggest that the association of Irf8 with Sting is induced early after HSV-1 infection.

6. *Along the same lines, does the IRF8 S151A mutant retain the ability to interact with STING in a manner inducible by cGAMP treatment?*

Reply: We have now performed new co-IP experiments as the reviewer suggested. The results showed that cGAMP stimulation enhanced the interaction between IRF8 and STING in overexpression system, whereas IRF8 S151A mutant lost the ability to interact with STING in untreated and cGAMP-treated cells (**new Fig. 5c, page 11**). These results suggest that S151 of IRF8 is important for its interaction with STING after cGAMP stimulation.

7. *Furthermore, the early effect of IRF8 KO on STING-induced IFN/ISG expression could potentially be explained by alteration of constitutive gene expression. RNAseq*

of unstimulated control versus KO cells could reveal whether IRF8 KO causes a more global alteration of genes related to STING signaling.

Reply: Following the reviewer's suggestion, we have performed RNA-seq to analyze a global alteration of genes related to cytosolic DNA-sensing pathway in WT and *Irf8*^{-/-} BMDMs. The GSEA showed that *Irf8*-deficiency had no significant effects on a global alteration of genes related to cytosolic DNA-sensing pathways in unstimulated cell (**new Fig. S1a, page 6**), suggesting that the role of *Irf8* in Sting-mediated signaling is not through its effects on basal expression of Sting-related genes

8. Where in the cell does IRF8 associate with STING? BFA treatment combined with co-IP or ImageStream analysis could be performed.

Reply: Our results suggest that *Irf8* was mostly co-localized with Sting in the ER (**Fig. 3g**). Following cGAMP stimulation, *Irf8* formed cytoplasmic punctate structures in the ERGIC and Golgi, where it was co-localized with S365-phosphorylated Sting (a hallmark of Sting activation) (**Fig. 3g-h**).

9. Figure 6a. The in vivo data may overestimate the effect of the described IRF8-STING mechanism on host defense, since IRF8 is involved in many other immunological processes of relevance for antiviral defense (as nicely described and cited by the authors; ref 44 reports on a role for IRF8 in control of the RNA virus LCMV). It would be nice to see whether Irf8^{-/-}Sting^{-/-} show the same phenotype as the single KO mouse strains. As a minimum, the authors should discuss this issue.

Reply: We would like to thank the reviewer for this insightful comment. We totally agree with the reviewer that the *Irf8*-Sting mechanism may not be responsible for all the *in vivo* effects of *Irf8*-deficiency on host defense, because of the fact that *Irf8* also plays important roles in other immunological processes of relevance for antiviral defense. However, our experiments in cellular and biochemical levels clearly demonstrated a role of *Irf8* in Sting- but not *Visa*/*Mavs*-mediated innate immune response in monocytes. We also showed that the role of *Irf8* in innate immune responses is independent of its role in differentiation of monocytes (Fig. S2). In addition, it has been previously shown that *Irf8*^{-/-} mice have intact host defense to the RNA virus VSV (Holtschke et al., Cell, 1996, PMID: 8861914), suggesting that *Irf8* does not have a general and dominant role in host defense. Therefore, our *in vivo* results with *Irf8*-deficiency mice support that the *Irf8*-Sting mechanism play a role in host defense against DNA virus. We have now added related discussion in the text (page 16-17).

It has been shown that expression of *Irf8* is restricted to monocytic cells, whereas Sting is ubiquitous expressed. In addition, recent studies (Wu et al., Immunity, 2020, PMID: 32640258; Yamashiro et al., Nat. Commun., 2020, PMID: 32636381; Gui et al., Nature, 2019, PMID: 30842662) reveal widespread IFN-independent activities of Sting in macrophages, which is independent of its phosphorylation at S365. In light of

these studies, we feel that it would be difficult to draw conclusions from additional studies with *Irf8*^{-/-}*Sting*^{-/-} mice.

Reviewer #2:

In this manuscript by Luo et al, the authors identified a new function of IRF8 in regulating STING signaling. IRF8 is a transcription factor playing important roles in development of myeloid cells especially DCs, NK cell activity, inflammasome activation, all of which acting through its transcriptional activity. The authors identified a new cytoplasmic activity of IRF8 in this manuscript that promotes STING signaling complex assembly and IFN signaling.

*The authors presented a thorough and convincing set of STING signaling assays clearly demonstrating that IRF8 is a positive regulator of the STING pathway. The authors pinned down the step where IRF8 acts (between TBK1 phosphorylation and STING polymerization and phosphorylation). In vivo HSV-1 infection and data on SLE and *Trex1*^{-/-} cells generally support the model.*

*How is IRF8 activated by DNA and how it is recruited to STING are not very clear. For example, based on published structural evidence, STING recruits TBK1 to one set of STING dimer and STING phosphorylation by TBK1 requires transacting on another set of STING dimers. The authors observed in *Irf8*^{-/-} cells pTBK1 but not pSting. Could IRF8 facilitate packing of STING dimers into polymers? The coIP studies in Fig 3E show substantial interaction at 9 hpi, which is pretty late for IRF8 to modulate STING activation. The authors should propose a reasonable model based on current understanding of the STING pathway, hopefully explain how/when DNA activates IRF8 phosphorylation, when does IRF8 interact with STING, how does that interaction impact TBK1 recruitment, STING polymerization and STING phosphorylation.*

Reply: We thank the reviewer for this very insightful comment. Our new endogenous Co-IP with BMDMs indicated that *Irf8* was weakly associated with *Sting* in uninfected cells, and their association was markedly increased at 2 hours after HSV-1 infection (**new Fig 3f**). In similar experiments, phosphorylation of *Tbk1*^{S172} occurred as early as 3 hours after HSV-1 infection (**new Fig. 4d, Fig. 2a**), while phosphorylation of *Sting*^{S365} or *Irf3*^{S388} occurred at 4 or 6 hours following HSV-1 infection (**Fig. 2a, new Fig. 4d**). Gel filtration experiments showed that *Irf8* and *Sting* were in the same complex in cells infected with HSV-1 for 4h (**Fig. 4c and S3b**). Confocal microscopy indicated that in rest cells, *Irf8* was mostly co-localized with *Sting* in the ER (**Fig. 3g**). Following cGAMP stimulation, *Irf8* formed cytoplasmic punctate structures co-localized with the ERGIC/Golgi, where it was co-localized with S365-phosphorylated *Sting* (**Fig. 3g-h**). In addition, our results also indicated that *Irf8*-deficiency impaired HSV-1-induced recruitment of *Irf3* to *Sting* but had no dramatic effects on recruitment of *Tbk1* to *Sting* (**new Fig. 4d**). *Irf8*-deficiency

reduced HSV-1-induced oligomerization of Sting but had no marked effects on its dimerization induced by HSV-1 (**Fig. 4a&b**). Taken together with other data in the paper and previous studies, we propose a model on how Irf8 modulates innate immune response to DNA virus.

Upon DNA virus infection, binding of cGas to viral DNA induces the synthesis of cGAMP, which binds to the ER-located Sting and promotes its dimerization. The dimerization of Sting causes its conformational changes, which facilitates the recruitment of Tbk1 and Irf8. In this complex, Tbk1 is autophosphorylated independent of Irf8, whereas Irf8 relieves the auto-inhibition of Sting and facilitates its polymerization. The polymerization of Sting conditions it for phosphorylation at S365 by Tbk1 and recruitment of Irf3. In the complex, Irf3 is subsequently phosphorylated by Tbk1, leading to its activation and induction of downstream antiviral genes. The recruitment of Irf8 to the Sting complex following DNA stimulation weakens its recruitment of the IKK complex and suppresses Sting-mediated NF- κ B activation, and has no effects on Sting-mediated autophagy. This has now been discussed in the text (**page 15**).

The authors showed increased NF κ B signaling (p-p65, p-I κ B) and decreased IFN signaling (pSTING and pIRF3) in IRF8^{-/-} cells when the STING pathway is activated by HSV-1 or VACV. If STING is not activated, how do they get NF κ B signaling? Both IFN and NF κ B are downstream signaling pathways that require STING activation, it is possible that IRF8 controls the balance between these two downstream pathways? They should activate STING with an agonist and measure IFN, NF κ B and LC3/autophagy to see if IRF8^{-/-} cells truly lost all STING activities.

Reply: We thank the reviewer for this insightful comment. In our experiments, we showed that HSV-1- and VACV-induced phosphorylation of mouse Sting^{S365} and Irf3^{S388} but not Tbk1^{S172} was inhibited in Irf8^{-/-} BMDMs (**Fig. 2a**). In addition, following HSV-1 infection, the recruitment of Tbk1 to Sting was barely affected in Irf8-deficient cells, whereas the recruitment of Irf3 to Sting was impaired in Irf8-deficient cells (**new Fig. 4d**). Based on our results, as well as various published studies, it is believed that Sting can mediate distinct downstream signaling pathways. We realize that it was somewhat misleading to generally state that Irf8-deficiency impairs Sting activation in our originally manuscript. We should have clearly stated that Irf8-deficiency affected the phosphorylation of Sting^{S365}, and its ability to recruit and activate Irf3 as well as induce transcription of downstream antiviral genes. In fact, it has been previously shown that phosphorylation of Sting^{S365} by Tbk1 is required for the recruitment and activation of Irf3 but does not affect Sting-mediated NF- κ B activation (Liu et al., Science, 2015, PMID: 25636800; Yum et al., PNAS, 2021, PMID: 33785602). This is consistent with our data showing that Irf8-deficiency impairs Sting-mediated Irf3 but not NF- κ B activation. In fact, as the reviewer pointed out, it is interesting that HSV-1 or VCAV-triggered NF- κ B signaling (p-p65, pI κ B) is even increased in Irf8-deficient cells. The simplest explanation is, as the reviewer suggested, that Irf8 acts as a modifier in monocytes by facilitating recruitment of Irf3

to S365-phosphorylated Sting while weakly suppresses the recruitment of the IKK complex to Sting.

Following the reviewer's suggestion, we have also measured cGAMP-triggered IFN, NF- κ B and autophagy signaling in *Irf8*^{-/-} cells. The results showed that *Irf8*-deficiency impaired cGAMP-induced phosphorylation of Sting^{S365} and Irf3^{S388} but not Tbk1^{S172} in BMDMs, whereas phosphorylation of p65 was increased in cGAMP-treated *Irf8*^{-/-} BMDMs (**new Fig. S1d**). Moreover, *Irf8*-deficiency had no marked effects on cGAMP-induced LC3 conversion (**new Fig. S1d**). These results again suggest that, *Irf8* is differentially required for Sting-mediated downstream effects. These results have been added to the revised manuscript. Based on our results, we have proposed a model on how *Irf8* modulates Sting-mediated innate antiviral response (see above), which has been discussed in the text (**see page 15**).

REVIEWERS' COMMENTS

Reviewer #1 (Remarks to the Author):

The authors have addressed the points that I raised, and I now find the conclusions well supported by the data.

Reviewer #2 (Remarks to the Author):

The authors have addressed all my concerns. The revised manuscript is much improved with more detailed mechanism. The IRF8 finding should be a novel mechanism for the STING biology as well as IRF8 biology.

Dear Editor and reviewers,

We would like to express our sincere gratitude to the reviewers for their constructive and positive comments again. Please see our point-by-point responses below.

Reviewer #1 (Remarks to the Author): The authors have addressed the points that I raised, and I now find the conclusions well supported by the data.

Response: We greatly appreciate your work in the improvement of our MS. Thanks again!

Reviewer #2 (Remarks to the Author): The authors have addressed all my concerns. The revised manuscript is much improved with more detailed mechanism. The IRF8 finding should be a novel mechanism for the STING biology as well as IRF8 biology.

Response: We greatly appreciate your work in the improvement of our MS. Thanks again!